# PAK4 suppresses RELB to prevent senescence-like growth arrest in breast cancer

Tânia D.F. Costa [1], Ting Zhuang[1,10], Julie Lorent[2], Emilia Turco[3], Helene Olofsson[1], Miriam Masia-Balague[1], Miao Zhao[1,11], Parisa Rabieifar[1], Neil Robertson[4], Raoul Kuiper [5], Jonas Sjölund[6], Matthias Spiess[1], Pablo Hernández-Varas[1], Uta Rabenhorst [1], Pernilla Roswall[7], Ran Ma[2], Xiaowei Gong [1], Johan Hartman [2], Kristian Pietras [6,7], Peter D. Adams [4,8,9], Paola Defilippi [3] & Staffan Strömblad[1]

Overcoming cellular growth restriction, including the evasion of cellular senescence, is a hallmark of cancer. We report that PAK4 is overexpressed in all human breast cancer sub-types and associated with poor patient outcome. In mice, MMTV-PAK4 overexpression promotes spontaneous mammary cancer, while PAK4 gene depletion delays MMTV-PyMT driven tumors. Importantly, PAK4 prevents senescence-like growth arrest in breast cancer cells in vitro, in vivo and ex vivo, but is not needed in non-immortalized cells, while PAK4 overexpression in untransformed human mammary epithelial cells abrogates H-RAS-V12-induced senescence. Mechanistically, a PAK4 – RELB - C/EBPβ axis controls the senescence-like growth arrest and a PAK4 phosphorylation residue (RELB-Ser151) is critical for RELB-DNA interaction, transcriptional activity and expression of the senescence regulator C/EBPβ. These findings establish PAK4 as a promoter of breast cancer that can overcome oncogene-induced senescence and reveal a selective vulnerability of cancer to PAK4 inhibition.

[1] Department of Biosciences and Nutrition, Karolinska Institutet, SE-141 83 Huddinge, Sweden. [2] Department of Oncology-Pathology, Karolinska Institutet, SE-171 77 Solna, Sweden. [3] Department of Genetics, Biology and Biochemistry, University of Torino, 10126 Torino, Italy. [4] Beatson Institute for Cancer Research, Bearsden, Glasgow G61 1BD, UK. [5] Department of Laboratory Medicine, Karolinska Institutet, SE-141 57 Huddinge, Sweden. [6] Division of Translational Cancer Research, Department of Laboratory Medicine, Lund University, SE-223 81 Lund, Sweden. [7] Division of Vascular Biology, Department of Medical Biochemistry and Biophysics, Karolinska Institutet, SE-171 77 Solna, Sweden. [8] Institute of Cancer Sciences, College of Medical, Veterinary and Life Sciences, University of Glasgow, Glasgow G12 8QQ, UK. [9] Sanford Burnham Prebys Medical Discovery Institute, 10901 North Torrey Pines Road, La Jolla, CA 92037, USA. [10] Present address: Henan Collaborative Innovation Center of Molecular Diagnosis and Laboratory Medicine, School of Laboratory Medicine, Xinxiang Medical University, Xinxiang 453003 Henan, P.R. China. [11] Present address: Department of Immunology, Genetics and Pathology, Uppsala University, SE-752 36 Uppsala, Sweden. Correspondence and requests for materials should be addressed to S.S. (email: staffan.stromblad@ki.se)

Breast cancer is the most common malignancy in women and causes significant lethality[1]. It is therefore imperative to find molecular actors that may be exploited as therapeutic targets.

Cellular senescence is a stress-inducible fail-safe mechanism that complements apoptosis in preventing the proliferation of damaged cells, while also contributing to ageing[2]. The senescent phenotype is highly heterogeneous and can be induced by different stimuli, including by oncogenes in untransformed cells (oncogene-induced senescence, OIS)[3]. Importantly, the identification of senescent cells in several pre-malignant or benign conditions suggests that senescence potently prevents cancer by acting as a major barrier to cancer development[4]. While no universal senescence-specific markers exist[5], senescent cells are characterized by a durable proliferative arrest often accompanied by increased senescence-associated β-galactosidase (SA-β-gal) activity and a senescence-associated secretory phenotype (SASP)[5] consisting of a plethora of secreted molecules that influence senescence through complex and poorly understood autocrine and paracrine means.

The nuclear factor kappa-light-chain-enhancer of activated B cells (NF-κB) acts a master regulator of the SASP[6]. NF-κB signaling is categorized into canonical (or classical) and non-canonical (or alternative) pathways, based on which components engage in the signaling cascade[7]. Canonical NF-κB signaling entails the processing of the precursor form NF-κB1 into the active form p50 and nuclear accumulation of RELA-p50 dimers. Distinctly, noncanonical NF-κB signaling involves the processing of NF-κB2 into p52 and assembly of RELB-p52 complexes[7]. While canonical NF-κB signaling (explicitly RELA) as well as CCAAT/enhancer-binding protein beta (C/EBPβ) are key transcriptional SASP regulators[8] and thereby firmly linked to cellular senescence, the potential involvement of noncanonical NF-κB signaling in senescence has remained largely unknown.

The serine/threonine p21-activated kinases (PAKs) are recognized for their privileged position at the intersection of major signaling pathways required for oncogenesis[9]. Among PAKs, PAK4 plays a pivotal role in various cancer-associated cellular events[9], which includes regulating different aspects of proliferation. In agreement, PAK4 is frequently overexpressed in cancer and correlates to poor patient prognosis in several cancer forms[10]. Yet, the potential functional role of PAK4 during cancer development in vivo has remained unknown and our understanding of the PAK4 signaling pathways that affect tumorigenesis is so far sparse.

Here we show that PAK4 is overexpressed in breast cancer associated with poor prognosis. PAK4 overexpression abrogates OIS in untransformed human mammary epithelial cells (HMECs), leads to mammary tumors in mice and suppresses senescence-like growth arrest in various breast cancer models, controlled by a PAK4–RELB–C/EBPβ regulatory axis. Our results reveal a critical function of PAK4 in breast cancer pathogenesis and identify a druggable vulnerability that may be exploited for breast cancer therapy.

## Results

### PAK4 expression is linked to breast cancer patient outcome.
The *PAK4* gene is located at a chromosomal region (19q13.2) frequently amplified in breast cancer with basal-like features[11] and consistently, PAK4 was found overexpressed in a small set of human breast cancer specimens[12]. In addition, we reported that high PAK4 levels correlate with poor survival of endocrine-treated breast cancer patients[13]. However, expression levels and copy number variation of PAK4 in relation to breast cancer patient outcome has not yet been examined in more general and larger sets of breast cancer patients. To this end, we analyzed the

METABRIC[14] dataset and found that PAK4 transcript expression was approximately twofold higher in breast tumors compared with their normal counterparts (Fig. 1a and Supplementary Fig. 1a). PAK4 mRNA levels were high across all breast cancer subtypes both when using the PAM50 signature[15] (Fig. 1b) and the IC10 classification[14] (Fig. 1c). The PAK4 overexpression in breast cancer relative to normal breast tissues was confirmed in two independent breast cancer datasets[16,17] (Supplementary Fig. 1b, c). PAK4 protein levels displayed a similar trend within a panel of six human breast cancer cell lines (Supplementary Table 1), most exhibiting PAK4 overexpression as compared with two independent batches of primary, non-immortalized HMECs (Supplementary Fig. 1d).

To analyze the clinical outcome of breast cancer patients in the METABRIC cohort, patients were stratified according to quartiles of PAK4 expression. Higher PAK4 expression was associated with worse disease-specific survival (DSS) in the entire cohort (Fig. 1d) as well as in patients that did not receive systemic adjuvant treatment (Fig. 1e). High expression levels of PAK4 also correlated with poor overall survival (OS) (Supplementary Fig. 1e). These conclusions withstand multivariate analyses, including lymph node status, breast cancer subtype, tumor size, and grade (Supplementary Tables 2 and 3).

PAKs overexpression in cancer varies widely and may be due to both mRNA upregulation and/or gene amplification[10]. PAK1 is the most amplified PAK in breast cancer (~8%), while PAK4 amplification is only detected in ~2% of breast tumors in The Cancer Genome Atlas (TCGA) cohort[10]. Using cBioPortal[18], we replicated this finding and also expanded the analysis to the METABRIC dataset, where we found a comparable fraction of tumors with PAK4 amplification (Supplementary Table 4). Interestingly, patients carrying tumors with PAK4 amplification tended to exhibit worse prognosis (Supplementary Fig. 1f, g). We also analyzed PAK4 copy number and mutational status in the breast cancer cell lines used throughout the study, but no relevant alterations were found (Supplementary Table 1).

Together, this indicates that PAK4 overexpression in breast cancer correlates with unfavorable disease outcome.

### PAK4 overexpression promotes mammary tumors.
While grafted immortalized mouse mammary epithelial cells overexpressing PAK4[19] and breast cancer cells with PAK4 depletion[20] shed some light on the potential relevance of PAK4 in breast cancer growth in vivo, the role of PAK4 during cancer development has not yet been examined.

To this end, transgenic MMTV–PAK4-overexpressing mice were generated in an inbred FVB/N strain (Supplementary Fig. 2a, b). Young MMTV–PAK4 mice were healthy, fertile, and had no overt phenotypic differences from wild-type (*wt*) mice. Importantly, from 6 months of age, PAK4-overexpressing virgin and nulliparous females exhibited lesions that eventually developed into mammary tumors in 25% of the cases (Fig. 2a, b). MMTV–PAK4 tumors morphologically resembled invasive lobular breast cancer and exhibited cribriform and fibrotic morphologies (Supplementary Fig. 2c). We also extracted genomic DNA from three mammary tumors that arose in virgin MMTV–PAK4 female mice (20–24 months old) plus liver tissue to conduct whole-exome sequencing (WES). The *Pak4* construct was visible in the sequencing coverage profiles as evident by very distinct exon/intron boundaries (Supplementary Data 1). Coverage for *Pak4* was about ten times higher than for genes surrounding the *Pak4* locus suggesting multiple integration sites. Interestingly, activating *Kras* mutations, including G12C and G13D, were found in two out of three MMTV–PAK4 tumors (Supplementary Data 1).

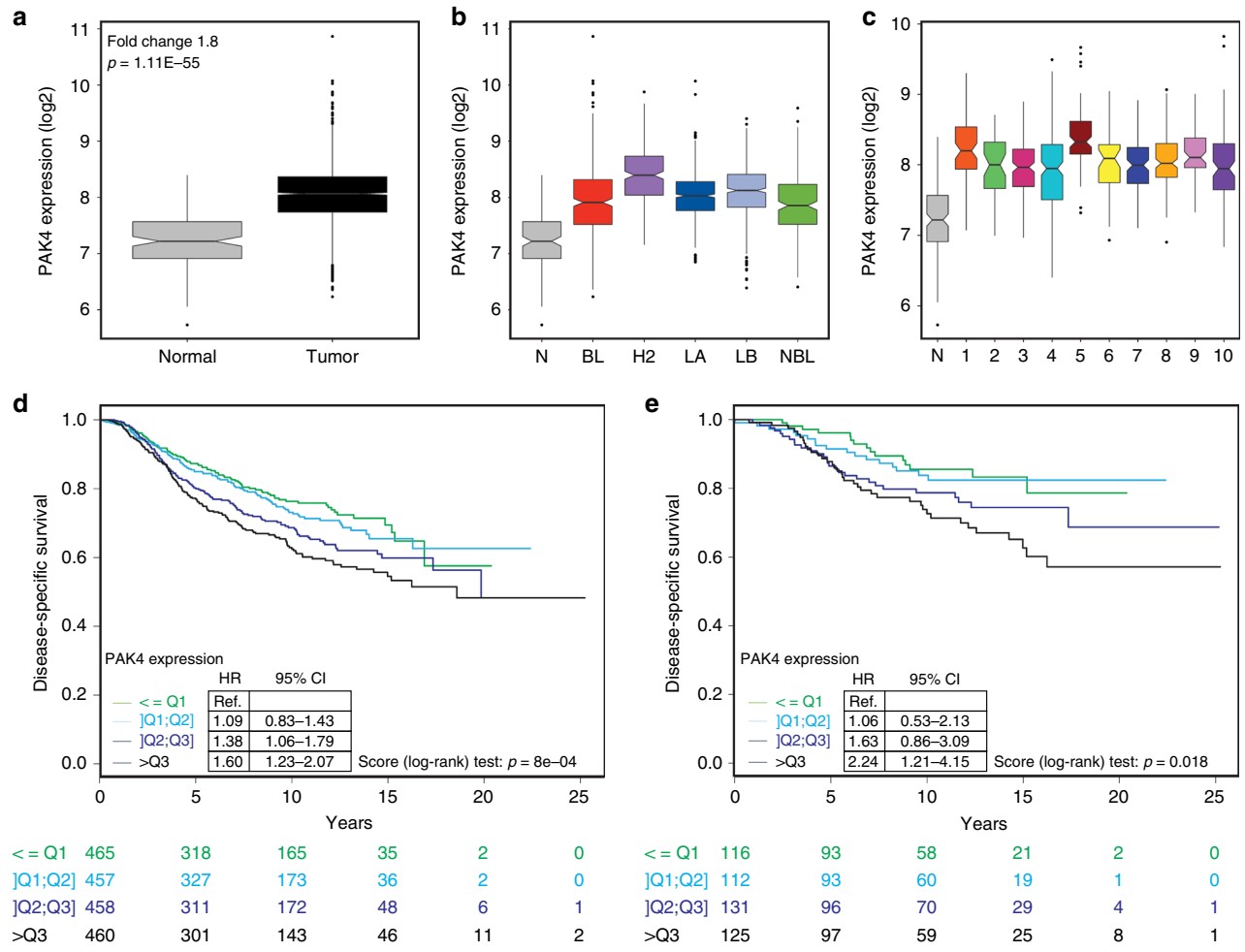

**Fig. 1** PAK4 overexpression in breast cancer is associated with unfavorable outcome. **a** PAK4 mRNA expression in breast carcinomas ($n = 1992$) and normal breast tissues ($n = 144$) in the METABRIC breast cancer cohort. **b** PAK4 expression across the five distinct molecular subtypes of breast cancer based on the PAM50 signature in the METABRIC breast cancer cohort (BL: Basal-like $n = 331$; H2: HER2-enriched $n = 240$; LA: Luminal A $n = 721$; LB: Luminal B $n = 492$; NBL: Normal Breast-like $n = 202$). **c** PAK4 expression among the ten distinct integrative clusters (IC) of breast cancer using the IC10 classification in the METABRIC breast cancer cohort (IC1 $n = 76$; IC2 $n = 45$; IC3 $n = 156$; IC4 $n = 167$; IC5 $n = 94$; IC6 $n = 44$; IC7 $n = 109$; IC8 $n = 143$; IC9 $n = 67$; IC10 $n = 96$). **d** Kaplan–Meier plot and analysis of disease-specific survival (DSS) in the METABRIC breast cancer cohort stratified according to quartiles of PAK4 expression. **e** Kaplan–Meier plot and analysis of DSS in the METABRIC cohort without systemic adjuvant treatment stratified according to quartiles of PAK4 expression. In **a**–**c** data are represented as boxplots where the middle line is the median, the lower and upper hinges correspond to the first and third quartiles, the upper whisker extends from the hinge to the largest value no further than $1.5 \times IQR$ from the hinge (where IQR is the interquartile range) and the lower whisker extends from the hinge to the smallest value at most $1.5 \times IQR$ of the hinge, while data beyond the end of the whiskers are outlying points that are plotted individually. The Fold Change and the $p$-value of a Mann–Whitney test are indicated in **a**. PAK4 expression in normal breast tissues (N) as seen in **a** is displayed to facilitate comparisons in **b** and **c**. The Hazard Ratios (HR), 95% confidence intervals (CI) and $p$-values by the log-rank (Mantel–Cox) test are indicated in **d** and **e**. $n$ is specified for each patient subgroup below **d** and **e**

**PAK4 depletion impairs PyMT-driven mammary tumorigenesis.** To further assess the function of PAK4 during mammary cancer development, we generated an additional genetically engineered mouse based on the widely used and relevant Polyoma Middle T (PyMT) transgenic breast cancer mouse model that is in part driven by PyMT-induced RAS signaling[21,22]. In this model, PyMT oncoprotein expression in the mammary epithelium is driven by the mouse mammary tumor virus (MMTV) promoter[21]. After a short latency, PyMT mice stochastically develop focal mammary tumors expressing biological markers consistent with those expressed in human breast cancers and that recreate stages similar to human breast cancer progression[22]. Importantly, PAK4 protein levels were elevated in PyMT-driven mammary tumors as compared with paired adjacent mammary tissues (Supplementary Fig. 2d–f). However, MMTV–PyMT tumors do not exhibit PAK4 amplification[23].

We crossed PAK4$^{fl/fl}$ mice[24] with MMTV-Cre mice/line D[25] to generate mice carrying conditional PAK4 depletion in the mammary epithelium (MMTV-Cre;PAK4$^{fl/fl}$, hereafter referred to as PAK4$^{MEp-/-}$ for simplicity). No significant defects in mammary gland morphogenesis were apparent in PAK4$^{MEp-/-}$ mice[26]. PAK4$^{fl/fl}$ mice were then interbred with MMTV-Cre and MMTV–PyMT mice to generate MMTV–PyMT;MMTV-Cre; PAK4$^{fl/fl}$ (PyMT;PAK4$^{MEp-/-}$) and MMTV–PyMT;MMTV-Cre (PyMT;PAK4$^{MEp+/+}$) control mice (Fig. 2c). PyMT-driven mammary tumorigenesis was examined in virgin females in the presence or absence of endogenous PAK4.

By 12 weeks of age, the PyMT oncogene induced extensive hyperplasia and mammary intraepithelial neoplasia in PyMT; PAK4$^{MEp+/+}$ control animals. In contrast, PyMT;PAK4$^{MEp-/-}$ glands were only mildly hyperplasic and exhibited fewer and smaller foci intermingled with normal ductal structures (Fig. 2d–e

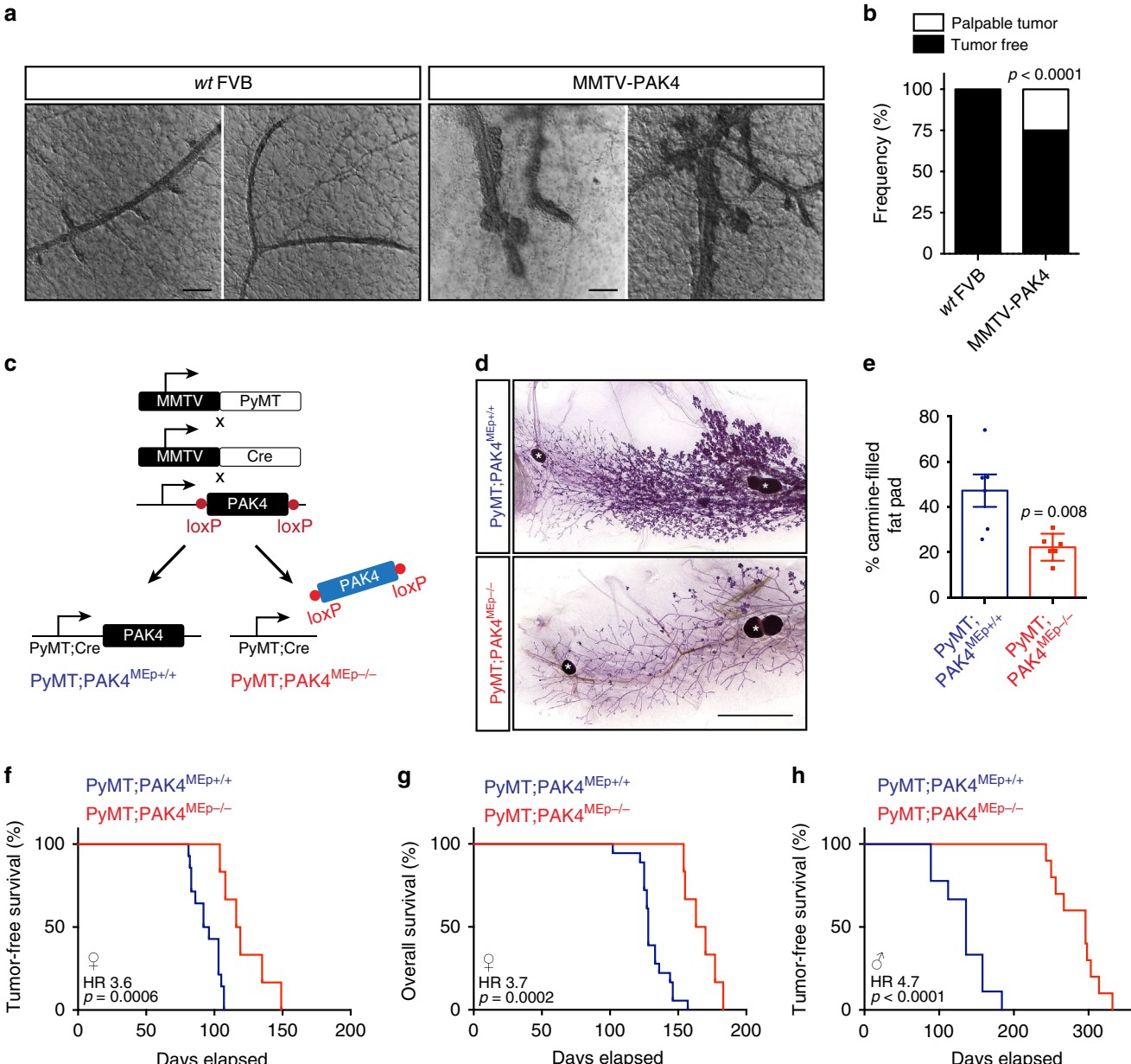

**Fig. 2** PAK4 promotes murine mammary tumorigenesis. **a** Representative images of mammary epithelium from 6-month-old females of the indicated genotypes. Scale bars, 200 μm. **b** Tumor incidence in aged virgin mice (20–24 months old) of the indicated genotypes ($n = 16$ *wt* FVB and $n = 20$ MMTV–PAK4). **c** Schematic diagram of the MMTV–PyMT mouse strains used. **d** Representative images of carmine-stained wholemounts of inguinal mammary glands from 12 weeks old female mice of the indicated genotypes. Scale bar, 5 mm. White asterisks denote lymph nodes. **e** Quantification of carmine-filled fat pad area ($n = 6$ per indicated genotype). **f** Kaplan–Meier plot of tumor-free survival in female mice of the indicated genotypes ($n = 14$ PyMT;PAK4$^{MEp+/+}$ and $n = 6$ PyMT;PAK4$^{MEp-/-}$). **g** Kaplan–Meier plot of overall survival in female mice of the indicated genotypes ($n = 18$ PyMT; PAK4$^{MEp+/+}$ and $n = 6$ PyMT;PAK4$^{MEp-/-}$). **h** Kaplan–Meier plot of tumor-free survival in male mice of the indicated genotypes ($n = 9$ PyMT; PAK4$^{MEp+/+}$ and $n = 10$ PyMT;PAK4$^{MEp-/-}$). Data are represented as percentage in **b** and as mean ± SEM in **e**. The *p*-values by chi-square test in **b** and by two-tailed unpaired *t* test in **e** are indicated. Dots in **e** depict individual samples. The Hazard Ratios (HR) and *p*-values by log-rank (Mantel–Cox) test are indicated in **f**–**h**

and Supplementary Fig. 2g). Consistently, PyMT;PAK4$^{MEp-/-}$ female mice developed palpable tumors significantly later than PyMT;PAK4$^{MEp+/+}$ mice, with the median tumor onset delayed on average by 22 days (Fig. 2f). This delay in tumor onset was also reflected in the median animal survival (Fig. 2g).

MMTV–PyMT also drives mammary tumor development in male mice, providing a different mammary tumor model with a considerably longer latency than in females. Strikingly, the effects of PAK4 gene depletion on tumor formation were even more prominent in the male PyMT (Fig. 2h).

Given the inherent caveats of the mouse strains herein employed[25], we evaluated the extent of PAK4 depletion in late stage mammary tumors that arose in females of both genotypes by measuring PAK4 protein levels by different techniques. While there was an overall consistent decrease in the levels of PAK4 in PyMT;PAK4$^{MEp-/-}$ tumors compared with controls, there was still substantial PAK4 expression in some PyMT;PAK4$^{MEp-/-}$ tumors, with significant variation between animals detected by immunoblot (Supplementary Fig. 2h, i). To analyze the spatial distribution of remaining PAK4 expression in the tumors, we

detected PAK4 mRNA levels using in situ hybridization (ISH/RNAScope). We found an overall lower PAK4 expression in early lesions compared with late-stage PyMT-driven tumors; that PAK4 expression is consistently less in the tissues harvested from PyMT;PAK4$^{MEp-/-}$ mice as compared with control; that residual PAK4 expression occurred specifically in tumor areas of PyMT;PAK4$^{MEp-/-}$ tissues (Supplementary Fig. 2j, k). This indicates that PAK4 was not completely depleted from the mammary tissues of PyMT;PAK4$^{MEp-/-}$ mice and PAK4 may therefore still contribute to tumorigenesis. ISH analysis of control tissues also points to the existence of small sub-populations of PyMT cells that express low or no PAK4. The behavior of such cells is likely not affected by Cre-mediated PAK4 depletion and hence, these cells may contribute as well to the tumor masses. In addition, we performed immunohistochemistry (IHC) with an anti-PAK4 antibody that we generated for this purpose (pab #73). The IHC results were in agreement with the other techniques (IB and ISH) used to evaluate PAK4 expression in these murine tissues (Supplementary Fig. 2l, m). The PAK4 labeling score correlated negatively with the mouse survival time (Supplementary Fig. 2n).

While several aspects might contribute to the variation of PAK4 expression observed, the most likely explanation stems from the previously recognized stochastic nature of transgene expression, including MMTV-Cre[25]. In our model, the expression of Cre recombinase and PyMT is not coupled within the exact same mammary epithelial cells, meaning that, stochastically, a mosaic pattern of transgene expression is expected, where some cells will co-express Cre and PyMT and other cells will only express Cre, PyMT, or none of the transgenes[25]. Due to the strong nature of the PyMT oncogene, it is likely that cells where only PyMT is expressed may eventually contribute to subpopulations of tumor cells, where PAK4 is not subjected to Cre-mediated knockout. To address this complexity, we have included IHC for Cre in tumors that arose in mice of both genotypes and scored the Cre expression (Supplementary Fig. 2o, p). A mosaic pattern of Cre expression was indeed present, particularly in PyMT;PAK4$^{MEp-/-}$ mice. A mosaic pattern of PyMT expression is less relevant here because only PyMT-transformed cells would develop into tumors. Interestingly, we observed a selectivity against Cre-positive cells in tumors that arose in PyMT;PAK4$^{MEp-/-}$ mice (Supplementary Fig. 2o, p), which likely underscores the here observed role of PAK4 in tumor development. This heterogeneity makes it challenging to relate PAK4 expression to other molecular events in this model. Nevertheless, our mouse model provides compelling evidence that mammary epithelial disruption of PAK4 results in impaired PyMT-induced tumorigenesis.

**PAK4 depletion induces senescence-like growth arrest**. To gain better understanding of the role of PAK4 in breast cancer, we performed RNA-Sequencing (RNA-Seq) of two human breast cancer cell lines, Hs 578T and BT-549, 72 h after transfection with PAK4-targeting siRNA. All replicates aligned well with >75% total mapping rate and a high level of concordance with paired reads (Supplementary Data 2). We observed a total of 500 versus 1151 genes that were differentially expressed (DE) following PAK4 knockdown in Hs 578T versus BT-549 cells, with 319 versus 702 genes downregulated and 181 versus 449 genes upregulated ($p < 0.05$ by Wald test with Benjamini–Hochberg false discovery rate correction and fold change ≥ 1.5) (Supplementary Fig. 3a, b and Supplementary Data 3 and 4). Importantly, PAK4 was significantly downregulated in both datasets (Supplementary Fig. 3a, b).

Ingenuity Pathway Analyses[27] reports were similar for both cell lines, revealing an overall prevalence of categories related to cell survival, growth, proliferation, and movement among the top

altered molecular and cellular functions upon PAK4 knockdown (Supplementary Fig. 3c, d).

Guided by the selectivity against Cre-positive cells observed in the tumors that arose in PyMT;PAK4$^{MEp-/-}$ mice; the Ingenuity Pathway Analyses upon PAK4 knockdown described above and an observation in glioblastoma[28], we analyzed potential senescence-like features in breast cancer cells upon PAK4 depletion. After transient transfection of two independent small interfering RNAs (siRNAs) targeting the human PAK4 gene, Hs 578T breast cancer cells adopted a flatter and larger senescence-associated morphology and exhibited elevated SA-β-gal activity (as measured with the two substrates X-Gal[29] and MUG[30]) that was accompanied by a significant decrease in BrdU-incorporation (Fig. 3a–d). Genes involved in cell cycle arrest, DNA damage/repair, and SASP factors are typically upregulated in senescent cells. Several of such genes were also upregulated upon PAK4 depletion in the RNA-Seq datasets and we validated a selection of these by RT-qPCR in independent RNA preparations of Hs 578T cells (Supplementary Fig. 3e). PAK4 knockdown also increased protein expression of the known senescence-regulators p53 and p21 (Supplementary Fig. 3f).

We next examined if PAK4 depletion induced senescence hallmarks in a diverse collection of cancer cells using several in vitro, in vivo, and ex vivo models. PAK4 was firstly abrogated in an extended panel of breast cancer cell lines BT-549, MDA-MB-231, MCF7, and T-47D (Supplementary Table 1). Increased SA-β-gal activity and decreased cell proliferation, both typical senescence responses, were observed upon PAK4 knockdown in all breast cancer cell lines here tested (Fig. 3f and Supplementary Fig. 3g, h). However, PAK4 knockdown in two independent batches of non-immortalized HMECs (#1 and #2) that express low levels of PAK4 (Supplementary Fig. 1d) did not induce SA-β-gal (Fig. 3f and Supplementary Fig. 3g). A primary cell line derived from a PyMT-driven tumor[31] also exhibited increased SA-β-gal activity when transfected with two independent siRNAs targeting murine PAK4 (Supplementary Fig. 3i). In addition, in contrast to untransformed HMECs, ex vivo bulk tumor cells derived from breast cancer patients (patient-derived cells, PDCs) arrested proliferation upon PAK4–siRNA treatment (Supplementary Fig. 3j). Expression of an siRNA-resistant PAK4 mutant restored the proliferation of breast cancer cells (Supplementary Fig. 3k), while the proliferation of breast cancer cells highly expressing kinase-dead EGFP–PAK4–M350 was impaired to an extent similar to that of siPAK4-transfected cells (Supplementary Fig. 3l). This suggests that PAK4-M350 acts as a dominant negative with respect to the studied phenotype. Moreover, breast cancer cells were generated using CRISPR/Cas9 genome editing with single guide RNAs (sgRNAs) targeting PAK4. The growth of sgPAK4 cells was dramatically inhibited as compared with parental cells, accompanied by increased SA-β-gal activity (Fig. 3g–j), consistent with the effect of siRNA-mediated PAK4 knockdown. Different PAK4-depleted breast cancer cell lines exhibited arrest in the G0/G1 or G2/M phases of the cell cycle (Supplementary Fig. 3m–o), as previously reported for cancer cells undergoing senescence[2].

Finally, we examined malignant cells of more diverse histological origins. PAK4 is overexpressed in pancreatic adenocarcinoma[10]. Consistently, PAK4 siRNA induced SA-β-gal in three out of four tested human pancreatic adenocarcinoma cell lines as well as in cells derived from the KPC transgenic mouse model of pancreatic adenocarcinoma[32] (Supplementary Fig. 3p, q). Given PAK4 overexpression in ovarian tumors[10], we also analyzed ovarian cancer cells and found that PAK4 knockdown induced SA-β-gal in OVCAR-3 cells, but not in TOV-21G or Caov-3 cells (Supplementary Fig. 3r). Thus, eleven out of fourteen cancer cell lines here examined exhibited increased SA-β-gal activity upon PAK4 knockdown.

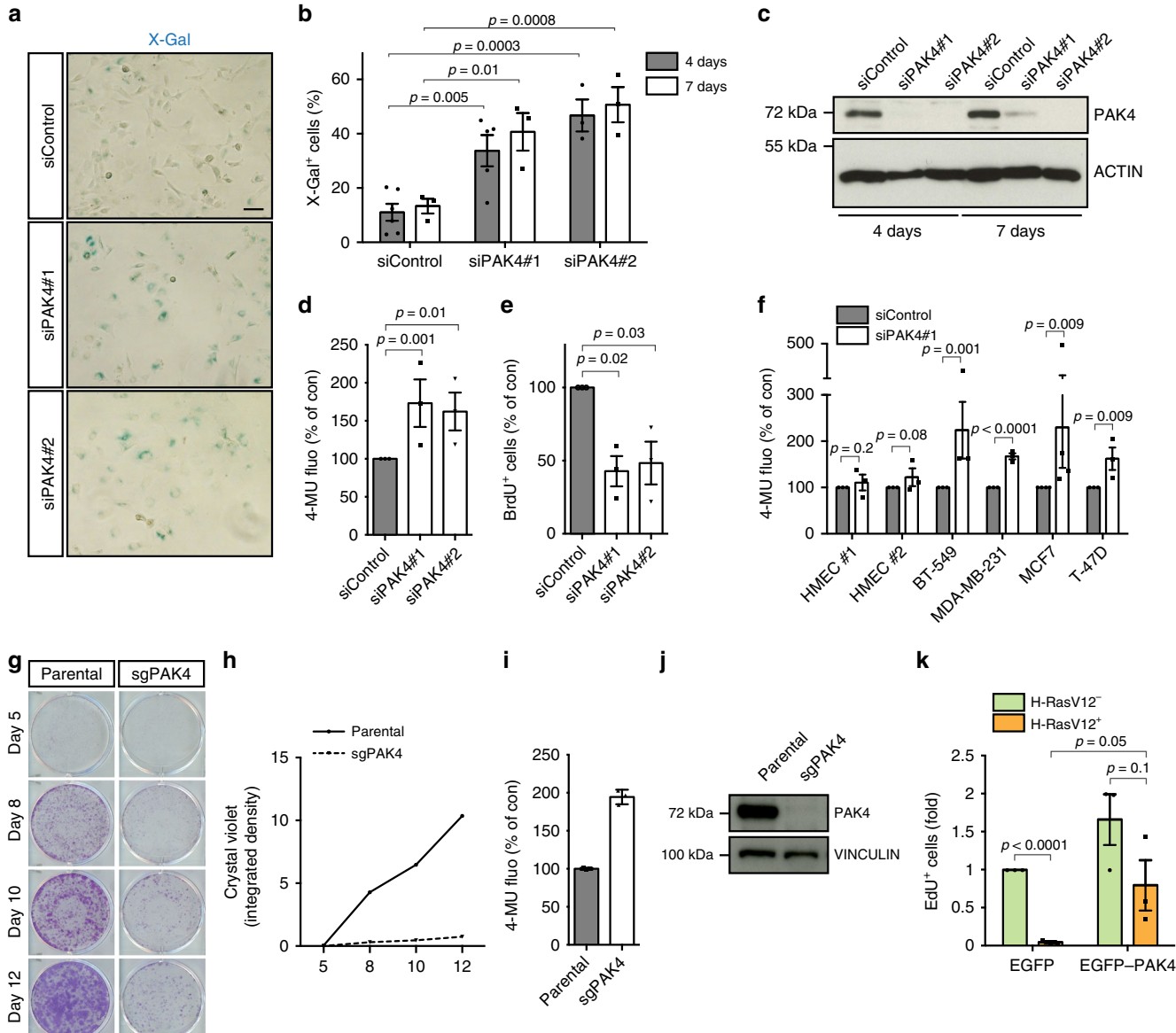

**Fig. 3** PAK4 depletion induces senescence-like growth arrest in cancer cells. **a** Representative images of X-Gal-stained Hs 578T cells, 4 days after transfection with the indicated siRNAs. Scale bar, 100 μm. **b** Quantification of SA-β-gal activity (% X-Gal+ cells) 4 days after transfection with siControl ($n = 6$), siPAK4#1 ($n = 5$), and siPAK4#2 ($n = 3$), and 7 days after transfection ($n = 3$ per group). **c** Representative immunoblot of Hs 578T cells, 4 and 7 days after transfection with the indicated siRNAs. **d** Quantification of SA-β-gal activity (4-MU fluorescence) in Hs 578T cells, 5 days after transfection with the indicated siRNAs ($n = 3$ per group). **e** Quantification of BrdU+ cells in Hs 578T cells, 5 days after transfection with the indicated siRNAs ($n = 3$ per group). **f** Quantification of SA-β-gal activity in a panel of breast cancer cells and two independent HMEC batches, 5 days after transfection with the indicated siRNAs ($n = 4$ for MCF7 and $n = 3$ for all other cells). **g** Representative images of colony formation assays of parental BT-549 and sgPAK4 cells generated by CRISPR/Cas9 at the indicated time points. **h** Growth curve of BT-549 cells generated with CRISPR/Cas9. **i** Quantification of SA-β-gal activity in BT-549 cells generated with CRISPR/Cas9. **j** Representative immunoblot of BT-549 cells after infection with the indicated sgRNAs. **k** Quantification of EdU+ cells in HMECs harboring inducible RAS^G12V 48 h after transient transfection with the indicated plasmids and simultaneous RAS^G12V induction ($n = 3$ per group). Data are represented as mean ± SEM in **b**, **d**, **e f**, and **k** and mean ± SD in **i**. *p*-values by two-way ANOVA in **b** and one-way ANOVA in **d** and **e** followed by Tukey's multiple comparisons *post hoc* test are indicated. The p-values by two-tailed unpaired *t* tests with correction for multiple comparisons using the Holm–Sidak method are indicated in **f** and **k**. Results in **g**, **h**, **i**, and **j** are representative of three independent transductions. Membranes in **c** and **j** were probed with antibodies against PAK4 using ACTIN or VINCULIN as a loading control

Taken together, this shows that inhibition of PAK4 triggers a senescence-like response in cancer cells of different histological origins.

**PAK4 overexpression overcomes H-RAS-V12-induced senescence.** Considering also that MMTV–PAK4 overexpression may have facilitated KRAS-driven mammary tumors (Fig. 2b and

Supplementary Data 1), and that oncogenic RAS-signaling in untransformed cells activates OIS[3], we tested the hypothesis that PAK4 may overcome oncogenic RAS-induced senescence. We utilized an established model where inducible H-RAS-V12 causes senescence-linked growth arrest in untransformed HMECs[33]. Indeed, while induction of H-RAS-V12 in this model induced an almost complete proliferation block, consistent with previous results from this model[33], overexpression of EGFP–PAK4

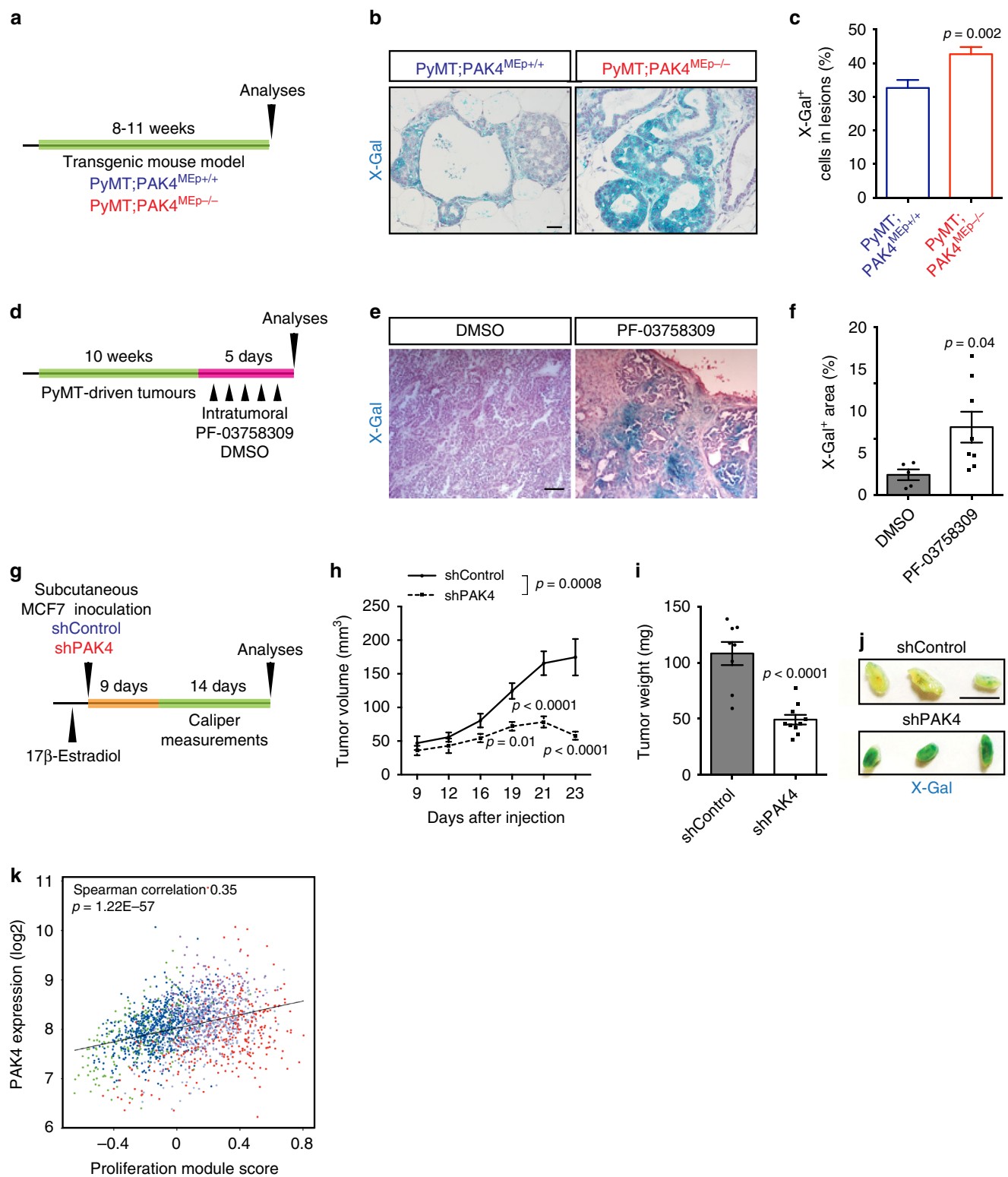

partially rescued proliferation (Fig. 3k). This suggests that PAK4 overexpression may overcome the OIS barrier to cancer.

**PAK4 inhibition induces senescence-like programs in vivo**. We then expanded our analyses to three distinct in vivo models. First, we analyzed X-Gal-stained mammary tissues from our model of

PyMT-driven tumorigenesis with conditional PAK4 knockout (Fig. 2 and Supplementary Fig. 2) harvested at 8–11 weeks of age. This time point was selected because senescent cells are abundant in the preneoplastic stage of certain cancers[4]. While most lesions displayed X-Gal positivity, we found an increase, albeit modest, in X-Gal positivity in PyMT;PAK4$^{MEp-/-}$ lesions as compared with control PyMT mice (Fig. 4a–c).

**Fig. 4** PAK4 inhibition induces senescence-like programs in vivo. **a** Schematic diagram of the experimental set up using the transgenic mouse model. **b** Representative X-Gal-stained PyMT-driven early mammary lesions from each indicated genotype. Scale bar, 100 μm. **c** Quantification of the percentage of X-Gal+ cells per lesion (PyMT;PAK4^MEp+/+ $n = 90$ and PyMT;PAK4^MEp−/− $n = 75$, entailing 15 lesions per animal derived from 5 to 6 animals per group). **d** Schematic diagram of the experimental set up using the PAK4 inhibitor PF-03758309. **e** Representative images of X-Gal-stained sections of PyMT tumors that were treated with the PAK4 inhibitor (PF-03758309) or DMSO. Scale bar, 250 μm. **f** Quantification of X-Gal+ areas in tumor sections of PyMT-treated tumors (DMSO $n = 5$ and PF-03758309 $n = 8$). **g** Schematic diagram of the experimental set up using xenografted MCF7 cells. **h** Kinetics of tumor growth for MCF7 cells stably expressing shPAK4 or shControl injected subcutaneously onto the back of nude mice (shControl $n = 8$ and shPAK4 $n = 10$). **i** Quantification of tumor weight at the experimental endpoint (shControl $n = 8$ and shPAK4 $n = 10$). **j** Representative X-Gal-stained wholemount tumor slices of the indicated groups. **k** Correlation between PAK4 expression and the proliferation score of breast tumors in the METABRIC breast cancer cohort ($n = 1992$). Data are represented as mean ± SEM in **c**, **f**, **h**, and **i**. p-values by two-tailed unpaired t test are indicated in **c**, **f**, and **i** and by two-way ANOVA (repeated measurements) followed by Bonferroni's multiple comparisons post hoc test in **h**. The Spearman correlation and p-value by Spearman's test are indicated in **k**. The colored dots represent the PAM50 subtypes as displayed in Fig. 1b

Next, we treated MMTV–PyMT mice with the PAK4 inhibitor PF-03758309[34]. Because PF-03758309 has been previously shown to block the growth of multiple human tumor xenografts[34], we used acute treatment where PF-03758309 was injected intratumorally for 5 days (Fig. 4d) followed by X-Gal staining. PF-03758309-treatment resulted in a higher proportion of X-Gal-covered tumor areas compared with vehicle-treated samples (Fig. 4e, f).

To further test the in vivo function of PAK4, MCF7 cells stably expressing shPAK4 or shControl (Supplementary Fig. 4a) were xenografted onto the back of nude mice. After injection and once the tumors were palpable (9 days post injection), we followed the detailed kinetics of tumor growth in vivo (Fig. 4g, h). shPAK4 tumors grew slower and were smaller at the experimental endpoint than control tumors (Fig. 4h, i and Supplementary Fig. 4b). Importantly, tumors expressing shPAK4 exhibited several hallmarks of senescence, including abundant SA-β-gal activity, decreased Ki67 positivity, and accumulated p53 (Fig. 4j and Supplementary Fig. 4c–e).

By performing analyses of gene expression data across the METABRIC cohort, we also found that higher expression of PAK4 was strongly correlated with known modules that reflect biological processes in breast cancer[35,36] (Supplementary Fig. 4f). Of relevance, we observed a positive correlation between PAK4 expression and the breast cancer proliferation score (Fig. 4k).

Together, these experiments extend our findings on the role of PAK4 to in vivo and patient settings.

**PAK4 inhibits NF-κB signaling**. To probe into the mechanism of PAK4 regulation of senescence-like growth arrest, we explored our RNA-Seq data of genes DE upon PAK4 knockdown. Interestingly, from both Hs 578T and BT-549 datasets we identified enrichment in NF-κB signatures, a crucial transcriptional regulator of the senescent phenotype[6], with a large number of NF-κB (RELA) target genes DE upon PAK4 knockdown (Fig. 5a and Supplementary Fig. 5a and Supplementary Data 5 and 6). Upregulation of several NF-κB target genes upon PAK4 knockdown was validated by RT-qPCR in Hs 578T cells, including the NF-κB subunits *NFKB1*, *NFKB2*, and *RELB* as well as the previously characterized NF-κB response genes *CD82*, *S100A4*, *TIMP2*, *CDKN1C*, *PRKCD*, *TWIST1*, *SPP1*, *TP53*, and *TRAF2* (Fig. 5b).

Importantly, we found that a metagene signature of NF-κB signaling[37] negatively correlated with PAK4 expression in the METABRIC breast cancer patient dataset, thus extending our findings into patient settings (Fig. 5c). Interestingly, this correlation was only apparent in breast tumors but not in normal breast tissue.

**PAK4 regulates senescence-like growth arrest via RELB**. To further investigate a potential relationship between PAK4 and NF-κB subunits in breast cancer, we performed correlation analyses for the expression of PAK4 and all NF-κB subunits in the METABRIC dataset. Considering expression as continuous variables, the expression of PAK4 and RELB displayed the strongest significant inverse association (Fig. 5d and Supplementary Fig. 5b). This inverse correlation was also observed at the protein level in Hs 578T breast cancer cells where PAK4 knockdown upregulated RELB (Supplementary Fig. 5c). In support of a potential clinical relevance, we also found that low PAK4/high RELB expression was associated with better prognosis in the HER2-positive breast cancer subtype (Supplementary Fig. 5d, e), in which PAK4 expression was the highest (Fig. 1b).

We then examined if the induction of growth arrest upon PAK4 knockdown in cancer cells functionally involved RELB. Indeed, RELB co-depletion with PAK4 restored the proliferative capacity of Hs 578T cells, defining RELB as essential for growth arrest upon PAK4 depletion (Fig. 5e and Supplementary Fig. 5c).

Thus, PAK4 is a negative regulator of NF-κB signaling in breast cancer cells, regulating senescence-like growth arrest via the noncanonical NF-κB subunit RELB.

**RELB-Ser151 phosphorylation promotes cell proliferation**. FLAG-tagged as well as endogenous PAK4 immunoprecipitated endogenous RELB, while FLAG-tagged RELB immunoprecipitated endogenous PAK4 (Fig. 6a and Supplementary Fig. 6a–c).

Interestingly, PAK4 strongly phosphorylated the REL-homology domain of RELB (RELB–RHD) (Fig. 6b, c and Supplementary Fig. 6d, e), consistent with NF-κB activity being controlled by phosphorylation[7]. Mapping Mass Spectrometry (MS) analysis identified serine 151 (Ser151) as a putative PAK4 site within RELB–RHD (Supplementary Fig. 6f). Phosphorylation of RELB–Ser151 is also indicated in the Phosida[38] database (Supplementary Fig. 6g). Importantly, phospho-null RHD-S151A (Ser→Ala substitution) abolished the PAK4 phosphorylation of the RHD, validating the MS results (Fig. 6c). Sequence alignment showed conservation of RELB–Ser151 across species (Fig. 6d).

In line with the notion that RELB signaling promotes growth arrest, overexpression of FLAG-tagged RELB *wt* led to reduced EdU incorporation in Hs 578T breast cancer cells, while expression of a FLAG-tagged phospho-mimicking RELB–S151E (Ser→Glu substitution) did not affect proliferation (Fig. 6e, f).

These results indicate that PAK4 is a RELB-kinase and that RELB phosphorylation on Ser151 is a novel mechanism controlling cell proliferation.

**RELB–Ser151 governs its DNA-binding and C/EBPβ transcription**. Interestingly, in the RELB–DNA complex crystal structure[39], Ser151 forms a hydrogen bond with the DNA backbone that, according to our computational modeling is abolished by Ser151 phosphorylation, thereby potentially weakening DNA binding (Fig. 7a). Importantly, this prediction was supported by our experiments, since the phospho-mimetic RELB–S151E had defective DNA binding in vitro as compared with *wt* RELB and the phospho-null RELB–S151A (Fig. 7b, c).

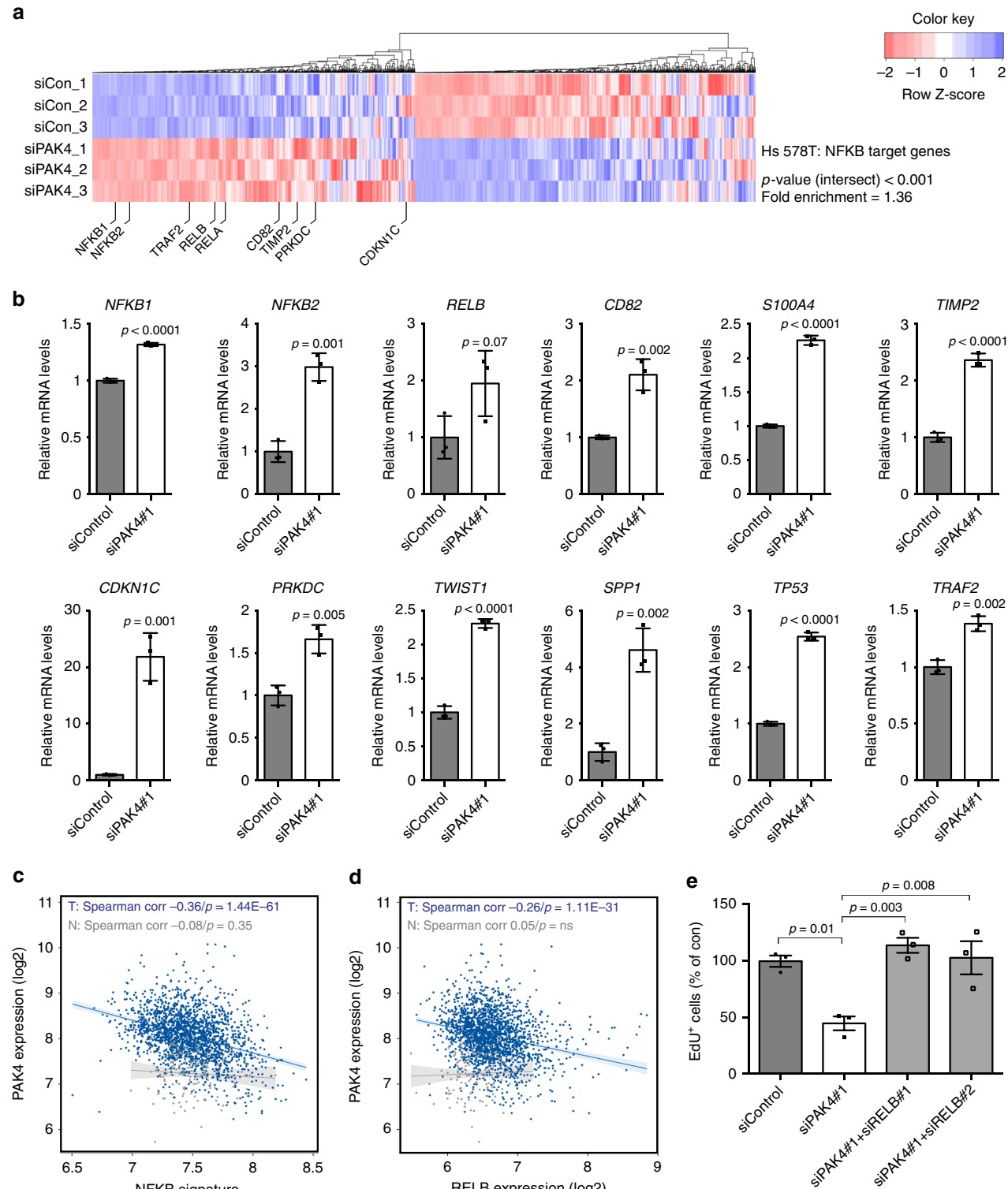

**Fig. 5** PAK4 inhibits senescence-like growth arrest via the NF-κB subunit RELB. **a** Column clustered heatmaps of NF-κB (RELA) target genes that are differentially expressed in Hs 578T cells upon siPAK4. Genes are by column and samples by row. The color intensity represents column Z-score, where red indicates highly and blue lowly expressed. Selected NF-κB target genes are indicated. **b** RT-qPCR validation of differentially expressed NF-κB target genes in Hs 578T cells. mRNA was prepared 72 h after transfection with the indicated siRNAs ($n = 3$ per group). **c** Correlation between PAK4 expression and an NF-κB signature in breast tumors (T) and normal tissue (N) in the METABRIC dataset. **d** Correlation between the expression of PAK4 and the alternative NF-κB subunit RELB in breast tumors (T) and normal tissue (N) in the METABRIC dataset. **e** Quantification of EdU+ cells in Hs 578T cells 4 days after transfection with the indicated siRNAs ($n = 3$ per group). Data are represented as mean ± SEM in **b** and **e**. $p$-values by two-tailed unpaired $t$ test are indicated in **b** and by one-way ANOVA followed by Tukey's post hoc test in **e**. The Spearman correlations and $p$-values by Spearman's test are indicated in **c** and **d** for tumors (T, blue, $n = 1992$) and normal tissue (N, gray, $n = 144$)

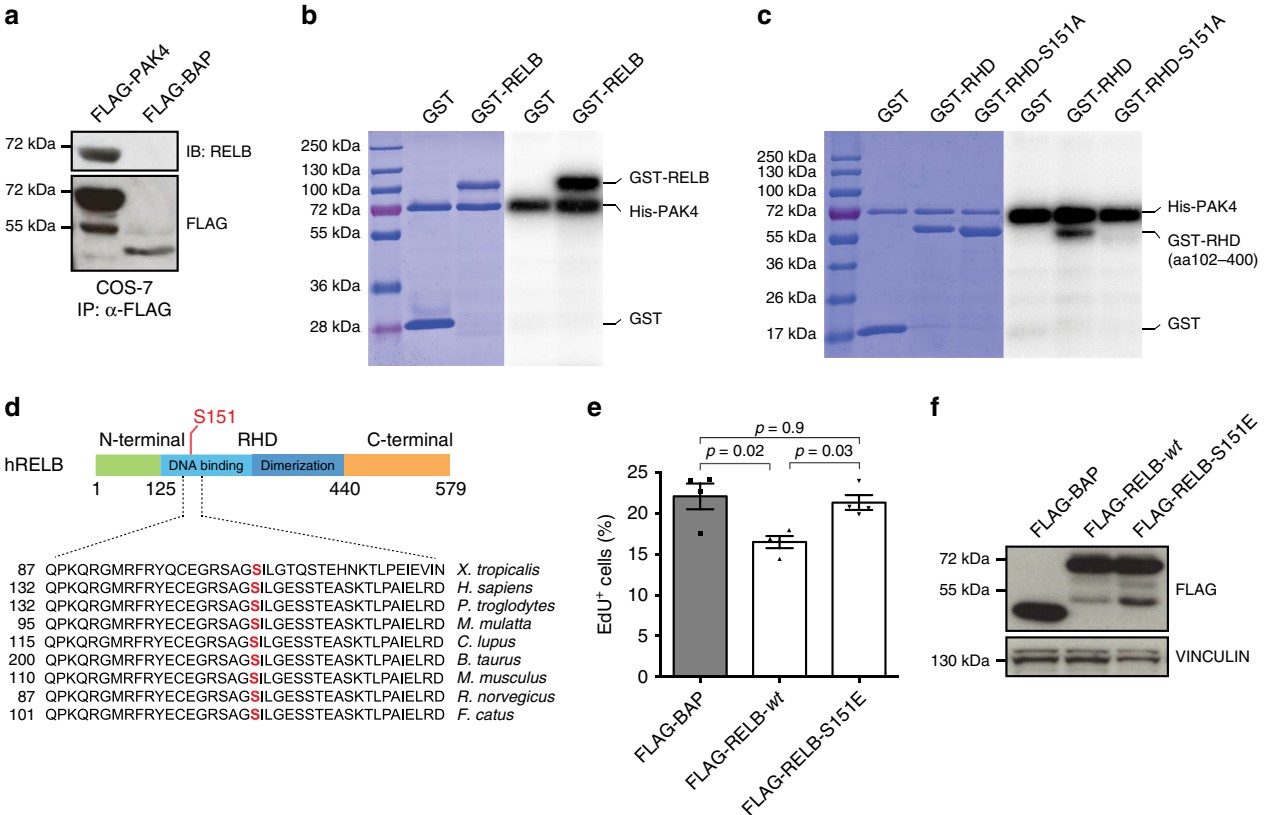

**Fig. 6** Phosphorylation of RELB-Ser151 promotes cell proliferation. **a** Immunoblot (IB) analysis for RELB and FLAG of FLAG-immunoprecipitates (IP) derived from COS-7 cells transfected with the indicated vectors. **b** Radioactive kinase assay with human recombinant His-tagged PAK4 and GST-tagged RELB full length. **c** Radioactive kinase assay using recombinant His-tagged PAK4 with GST, GST-tagged REL-homology domain of RELB (GST-RHD) or non-phosphorylatable GST-RHD-S151A. **d** Sequence alignment of Ser151 present in human RELB across different species. **e** Quantification of EdU$^+$ cells in Hs 578T cells 36 h after transient transfection with the indicated constructs ($n = 4$ per group). **f** Representative immunoblot of protein overexpression for experiments quantified in **e**. Data are represented as mean ± SEM in **e** and the $p$-value by one-way ANOVA followed by Tukey's post hoc test is indicated. Results in **a**–**c** are representative of three independent repeats. Coomassie-stained gels are shown to the left as loading control in **b** and **c**

Consistently, PAK4 overexpression in breast cancer cells inhibited RELB binding to DNA, while PAK4 depletion increased RELB DNA-binding (Fig. 7d, e and Supplementary Fig. 7a, b). To examine Ser151-phosphorylation effects on RELB transcriptional activity, we overexpressed FLAG-tagged RELB *wt* and RELB–S151E and compared, by qPCR array, the expression of candidate genes from our lists of DE genes upon PAK4 knockdown derived by RNA-Seq (Supplementary Fig. 7c and Supplementary Data 3 and 4). Interestingly, C/EBPβ, an important senescence regulator[8], emerged as the top altered candidate (Fig. 7f and Supplementary Fig. 7c). In agreement, we found a positive correlation between RELB and C/EBPβ expression in the METABRIC patient dataset (Fig. 7g). Further, PAK4 silencing increased C/EBPβ expression (Fig. 7h) and co-depletion of C/EBPβ with PAK4 partially rescued the inhibited proliferation induced by PAK4 knockdown (Fig. 7i and Supplementary Fig. 7d).

These data demonstrate that phosphorylation of Ser151 abrogates RELBDNA-binding and consequently negatively influences RELB-mediated transcription (i.e., C/EBPβ expression) impacting cell proliferation (Fig. 7j).

## Discussion

Here we identify PAK4 as an attractive breast cancer drug target candidate given its overexpression in breast cancer patients negatively affecting prognosis and its key function in evading senescence-like growth arrest in breast cancer cells via the non-canonical NF-κB component RELB.

The lack of investigations on the potential role of PAK4 in tumorigenesis in vivo prompted us to establish two transgenic models, with PAK4 overexpression and depletion, respectively, specifically in the mouse mammary gland. PAK4 *wt* over-expression caused hyperplasia and, at a later stage, mammary tumors in a fraction of the mice comparable to the overexpression of catalytically active PAK1[40]. This incomplete penetrance as well as the late tumor onset may suggest that similarly to PAK1, PAK4 itself acts as a relatively weak oncogene and/or to facilitate tumor formation driven by other oncogene(s), the latter supported by the activating KRAS mutations that we observed in these tumors and the finding that PAK4 overexpression overcame RAS-induced senescence in mammary epithelial cells.

The PAK4 conditional depletion was made in the context of PyMT-driven mammary tumorigenesis. PyMT mice lacking PAK4 developed mammary tumors significantly later than control mice. This was likely a consequence of the observed reduced burden of hyperplasia and pre-malignant lesions detected in early stage glands of PAK4-depleted animals. Also this impaired PyMT-driven tumor initiation is consistent with a role for PAK4 in senescence evasion, an early event in tumorigenesis. This notion is corroborated by our observations of a restored senescence-like response in PAK4-depleted breast cancer xeno-grafts and in PyMT tumors with conditional PAK4 knockout and PyMT tumors treated in vivo with a PAK4 inhibitor that has

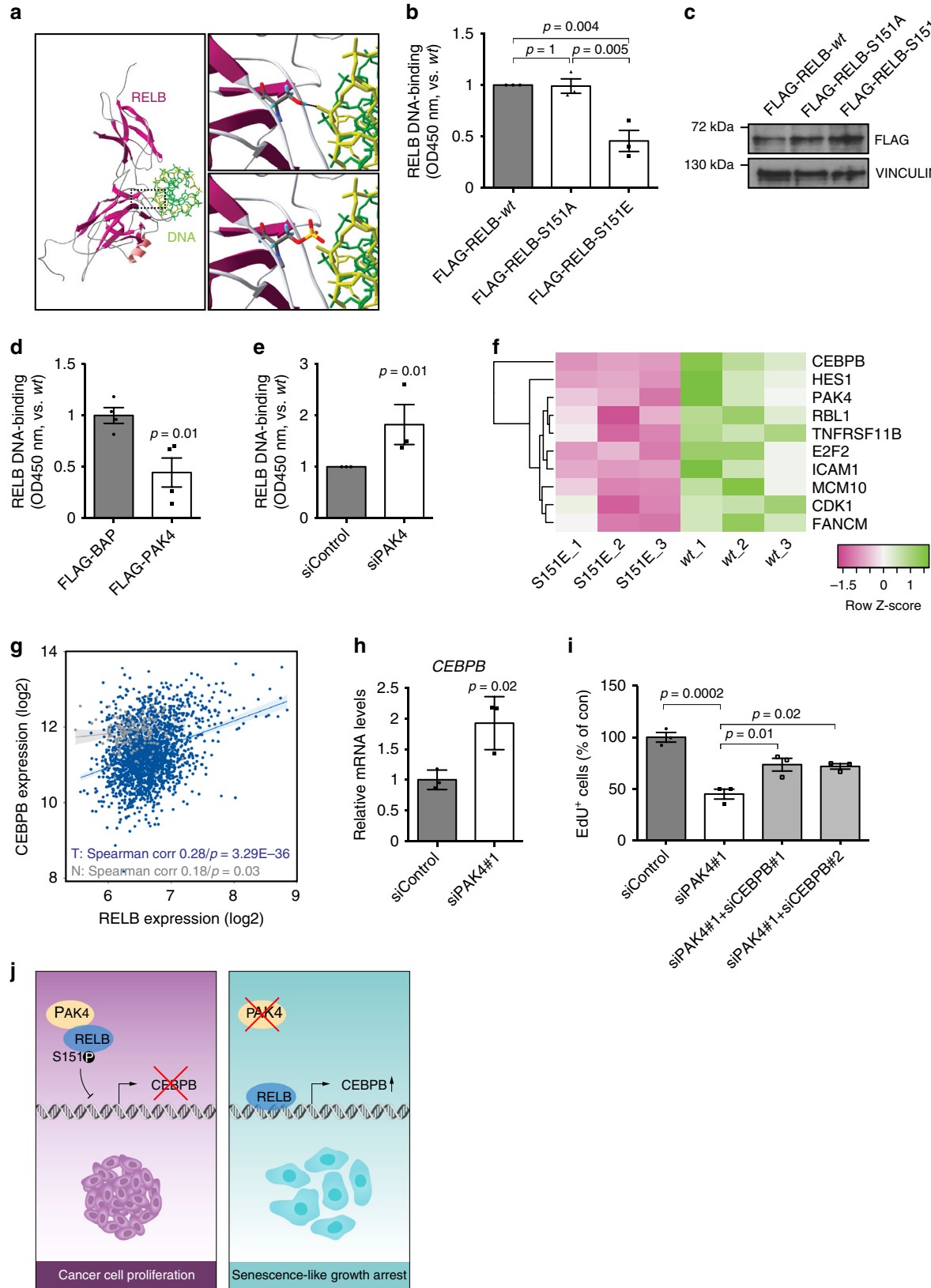

proved successful in blocking the growth of multiple human tumor xenografts[34].

The restored senescence-like response in vivo upon PAK4 inhibition is supported by extensive in vitro phenotypic characterization and transcriptome analyses employing a wide variety

of tools (siRNAs, shRNA, CRISPR/Cas9, and protein over-expression). It is also in agreement with earlier reports showing that PAK4 promotes cell cycle progression when overexpressed, and hinders such progression when removed or inhibited[19,20]. Beyond that, our data show that PAK4 inhibition almost

**Fig. 7** RELB–Ser151 phosphorylation controls DNA-binding and C/EBPβ transcription. **a** Structure of RELB:κB DNA complex (PDB ID: 3DO7) highlighting RELB–Ser151 in close proximity to DNA, the RELB subunit shown in purple and DNA in greens. Top inset zooms into Ser151 and shows a hydrogen bond with the DNA backbone. Bottom inset models the disturbing impact on DNA binding of Ser151 phosphorylation. **b** ELISA-based measurements of RELB DNA-binding 24 h upon expression of the indicated constructs in Hs 578T cells ($n = 3$ per group). **c** Representative immunoblot for experiments quantified in **b** using VINCULIN as loading control. **d** ELISA-based measurements of RELB DNA-binding in MCF7 cells stably expressing the indicated constructs ($n = 4$ per group). **e** ELISA-based measurements of RELB DNA-binding 4 days after the indicated siRNA transfections in Hs 578T cells ($n = 3$ per group). **f** Differentially expressed genes from an RT-qPCR array performed in Hs 578T cells transfected with FLAG-tagged RELB–S151E and RELB-*wt* ($n = 3$ per group). Statistically discernible hits (Supplementary Fig. 7c) are ordered by Fold Change. **g** Correlation between the expression of C/EBPβ and RELB in tumors (T) and normal tissue (N) in METABRIC. **h** RT-qPCR analyses of C/EBPβ mRNA in Hs 578T cells 72 h after the indicated siRNA transfections ($n = 3$ per group). **i** Quantification of EdU$^+$ cells in Hs 578T cells 4 days after the indicated siRNA transfections ($n = 3$ per group). **j** Schematic summary diagram. In proliferating cancer cells, abundant PAK4 contributes to senescence avoidance. PAK4 inhibits RELB by direct interaction and phosphorylation of RELB–Ser151 thereby impairing binding affinity to DNA. In the absence of PAK4, RELB accumulates and binds DNA, which culminates in the expression of senescence regulators including C/EBPβ. Data are represented as mean ± SEM in **b**, **d**, **e**, **h**, and **i**. *p*-values by one-way ANOVA followed by Tukey's post hoc test in **b** and by two-tailed unpaired *t* tests in **d**, **e**, **h**, and **i** are indicated. The Spearman correlation and *p*-value by Spearman's test are indicated in **g** for tumors (T, blue, $n = 1992$) and normal tissue (N, gray, $n = 144$)

invariably arrests cancer cell proliferation, while inducing additional senescence-like features (including morphological changes, increased SA-β-gal activity, and gene expression changes indicative of a senescent-like phenotype). Our data thus suggest that, apart from glioblastomas[28], epithelial cancer cells of various origins require PAK4 to avoid senescence, adding to the idea of generalized PAK4 addiction in cancer[9].

NF-κB signaling has diverse and complex roles in cancer with reports that NF-κB seems to both contribute to promotion and suppression of tumorigenesis depending on the cellular context[7]. However, while canonical NF-κB signaling (essentially via RELA) has been extensively associated with the senescent phenotype[8,41], the potential role in senescence for the noncanonical NF-κB pathway has remained elusive[42]. We identify PAK4 as an inhibitor of NF-κB signaling in breast cancer and show that the senescence-like growth arrest upon PAK4 knockdown in cancer cells functionally requires the noncanonical NF-κB subunit RELB. RELB is a known transcriptional target of both types of NF-κB signaling[7], meaning that the increased mRNA levels of NFKB1, NFKB2, and RELB, as well as the activation of RELB could act in synergy with increased RELB levels upon PAK4 depletion. Furthermore, in line with recent studies that indicate that posttranslational modifications of NF-κB subunits play a critical role in fine-tuning transcriptional activity, we identified PAK4 as a RELB kinase. Adding to the four previously validated RELB phosphosites[43], we identified Ser151 as a PAK4-site within the RHD of RELB, whose phosphorylation status regulated the RELB binding to DNA and consequently, RELB transcriptional activity. RELB–Ser151 phosphorylation prevented RELB interaction with DNA possibly due to electrostatic repulsion with the DNA phosphate backbone[39]. Such a modification may not only affect the binding of different homo- or heterodimers of NF-κB to cognate κB sites, but also regulate DNA-binding and release, thereby fine-tuning transcription. It may also affect the recruitment of various transcriptional coactivators and/or repressors to the promoters of NF-κB target genes. The RELB–Ser151 PAK4 phosphorylation site also affected the expression of C/EBPβ, whose expression is critical for senescence[8]. Growth arrested PAK4-depleted cells show increased binding of RELB to DNA consistent with a previous report of RELB–DNA-binding activities during therapy-induced senescence[44]. Interestingly, NF-κB (explicitly RELA) and C/EBPβ can cooperatively regulate the SASP[8]. Here we present a direct link between RELB and C/EBPβ, raising the interesting question that these previously considered distinct pathways may be regulated in an integrated manner to control cell fate.

Further raising the potential clinical relevance, we found that PAK4 is overexpressed across all breast cancer subtypes,

specifically upregulated during the disease development and correlated with poor prognosis. In addition, we found that the small fraction of patients carrying tumors that harbor PAK4 amplification tend to exhibit worse prognosis, suggesting that although not a very frequent alteration, PAK4 amplification may still be a clinically relevant feature. Importantly, we found that a metagene signature of NF-κB signaling[37], as well as RELB expression itself, negatively correlated with PAK4 expression in the METABRIC breast cancer patient dataset. These correlations were only apparent in breast cancer but not in normal breast tissue, adding to the notion that it may be possible to selectively target the addiction to PAK4 of cancer cells. In addition, we show that the low PAK4/high RELB pattern of expression correlates with better prognosis in HER2-positive breast cancer patients. This is consistent with the observation that HER2-positive patients exhibited the highest PAK4 expression among the PAM50 and IC10 breast cancer subtypes and also consistent with the strong correlation observed between PAK4 expression and HER2 signaling in the METABRIC dataset. PDCs were also sensitive to PAK4 abrogation ex vivo while primary non-immortalized HMECs were not, reemphasizing the notion of PAK4 as a potential breast cancer treatment target.

In summary, our findings establish PAK4 as a promoter of breast cancer development, possibly through overcoming the barrier of OIS. By showing also a competitive advantage conferred by PAK4 to established breast cancer cells, acting through the noncanonical NF-κB signaling subunit RELB to prevent senescence-like growth arrest, we reveal a selective vulnerability of cancer cells to PAK4 inhibition that may be explored as a therapeutic strategy.

## Methods
**Mice**. All experimental animal procedures in this study were performed in accordance with Swedish and European Union guidelines and approved by the institutional ethical commissions (Stockholms Södra and Linköpings djurförsöksetiska nämnder S39-09; S169-09; S86-11; S23-12; S169-12; S66-14 and 42-16).

Animals were housed in group in standardized cages with a 12:12 h light:dark cycle with unrestricted access to food and water. At the experimental endpoint, animals were sacrificed by cervical dislocation after isoflurane anesthesia. Both males and females were used as specified in each section.

To generate the MMTV-myc-PAK4 sequence, we amplified by PCR the coding sequence of mouse PAK4 from a commercial cDNA clone IRAVp968C0391D (Sourcebiosciences) and cloned the amplified cDNA in pcDNA vector containing myc tag sequence at the 5′ end of the multiple cloning site. Then, we isolated the myc-PAK4 fragment by digestion with KpnI and NotI, and inserted it into the pspT2 MMTV–LTR plasmid (a kind gift from M. Glukova, Paris). The MMTV-myc-PAK4 sequence was isolated from the vector by digestion with SalI, gel purified, and microinjected at 3.4 ng/μL in the pronucleus of fertilized oocytes from FVB/NJ mice (Charles River) according to standard protocols. MMTV–PAK4 is available upon request for noncommercial use.

Transgene integration was tested via PCR analysis of genomic DNA, with the MMTV–PAK4 primers 5′-TTTGGAAAGAAGAAGAAGC-3′ and 5′-ACTGCC ACTGCCTGCCTCAC-3′, amplifying a band of 441 bp.

A cohort of 36 virgin females ($n = 16$ wt FVB and $n = 20$ MMTV–PAK4) was monitored for the development of mammary tumors up to 24 months of age to determine tumor incidence.

PAK4[fl/fl] mice (B6.129S2(FVB)-Pak4[tm2.1Amin]/J, gift from Audrey Minden, Rutgers University, NY, USA)[24] were crossed with MMTV–Cre/Line D mice (Tg(MMTV-cre)4Mam/J, Jackson Laboratory)[25] and then bred to MMTV–PyMT (B6.FVB-Tg(MMTV-PyVT)634Mul/LellJ, gift from Lars Holmgren, Karolinska Institutet, Stockholm, Sweden)[45]. All mice were maintained on B6 background.

Cohorts of virgin MMTV–PyMT;MMTV-Cre;PAK4[fl/fl] (PyMT;PAK4[MEp−/−]) and MMTV–PyMT;MMTV-Cre (PyMT;PAK4[MEp+/+]) were used to study the effects of conditional PAK4 depletion in mammary tumorigenesis. For tumor-free survival analyses female mice were monitored for tumors by weekly palpation after the age of 2 months. The endpoint for survival analysis was taken as the time at which the combined tumor burden surpassed the ethical guidelines in our permit.

The presence of the PyMT and Cre transgenes and the heterozygous or homozygous knockout of Pak4 was verified through genotyping. Genomic DNA was prepared from either ear or tail biopsies using the fast tissue–to-PCR kit (#K1091, Fermentas) and PCRs ran according to the Jackson Laboratory suggested protocols. Genotypes were visualized on 1–2.5% agar gels stained with GelRed (41003, Biotium). The primer pairs used (synthesized by ThermoFisher) were as follows: Pak4: 5′-CGGATATTGTCACCCACACCA-3′ and 5′-CTAACAGGGACA GGAGCT-3′; Cre: 5′-GGTTTCCCGCAGAACCTGAAG-3′ and 5′-GCTAGTGCC TTCTCTACACC-3′; and PyMT: 5′-GGAAGCAAGTACTTCACAAGGG-3′ and 5′-GGAAAGTCACTAGGAGCAGGG-3′.

Female 10–15-week-old MMTV–PyMT (FVB/N-Tg(MMTV–PyVT)634Mul/J)[21] were used to evaluate PAK4 expression in mammary tumors and adjacent mammary tissue and to study the intratumoral effects of PF-03758309.

Female 6–8-week-ld BALB/c mice (CAnN.Cg-Foxn1[nu]/Crl, Charles River) were used in the xenograft model. BALB/c mice acclimatized for 1 week before being randomly assigned per experimental condition.

**Cell lines**. HMEC (batches #1 and #2), Hs 578T, BT-549, MCF7, MDA-MB-231, T-47D, OVCAR-3, Caov-3, and TOV-21G were obtained from the American Type Culture Collection (ATCC). SUM-159 cells were obtained from Asterand. AsPC-1, PANC-1, and MIA Pa-Ca-2 were obtained from the European Collection of Authenticated Cell Cultures. These pancreatic cancer cells and Paca3[46] were a generous gift from Rainer Heuchel and Matthias Löhr, Karolinska Institutet, Huddinge, Sweden. Cells were tested for mycoplasma but not subjected to additional cell authentication. HMECs were grown in Mammary Epithelial Cell Growth Medium (MEGM, CC-3051, Lonza) supplemented with MEGM BulletKit (CC-3150, Lonza). Hs 578T, MCF7, MDA-MB-231, PANC-1, and MIA Pa-Ca-2 cells were grown in Dulbecco's Modified Eagle's Medium that contains 4,5 g/L glucose and 4 mM L-glutamine (DMEM, 41965, Life Technologies) supplemented with 10% Fetal Bovine Serum (FBS, 10270, Life Technologies). BT-549, T-47D, AsPC-1 and Paca3 cells were grown in RPMI-1640 (42401, Life Technologies) supplemented with 2 mM L-glutamine (25030, Life Technologies) and 10% FBS. OVCAR-3 cells were grown in RPMI-1640 supplemented with 2 mM L-glutamine, 20% FBS, and 0.01 mg/mL bovine insulin (I5500, Sigma). Caov-3 were grown in DMEM supplemented with 10% FBS and 1 mM sodium pyruvate (11360, Life Technologies). TOV-21G were grown in a 1:1 mixture of MCDB 105 medium (M6395, Sigma) and Medium 199 (M0650, Sigma) supplemented with 15% FBS. SUM-159 cells were grown in Ham's F-12 Nutrient Mix (11765, Life Technologies) supplemented with 5% FBS, 10 ng/mL epidermal growth factor (EGF, PHG0311, Life Technologies), 5 μg/mL insulin (I5500, Sigma), 1 μg/mL hydrocortisone (H0888, Sigma), and 10 mM HEPES (15630, Life Technologies). HMECs harboring 4-Hydroxytamoxifen (OHT)-inducible RAS[G12V][32], kindly provided by David Beach, Barts and The London School of Medicine and Dentistry, UK, were grown in phenol red-free Mammary Epithelial Basal Medium (CC-3153, Lonza) supplemented with MEGM BulletKit (CC-3150, Lonza). These cells display a classical senescence phenotype upon treatment with 12.5 nM OHT (H7904, Sigma). MCF7 cells stably expressing shControl or shPAK4 have been previously described[47] and were grown in DMEM supplemented with 10% FBS and 150 μg/mL G418 (11811, Life Technologies). Stably-transfected MDA-MB-231-derived cell lines were generated by electroporation using the Neon Transfection System (ThermoFisher). Briefly, EGFP, EGFP–PAK4-wt, and EGFP–PAK4–siRNA resistant constructs and the pGL4.21 vector (E6761, Promega) containing a Puro[R] cassette were co-transfected using the electroporation parameters as suggested by the manufacturer. After 48 h recovery in complete medium, cells were seeded at low density in medium containing 2 μg/mL puromycin (P8833, Sigma). This dose was effective for killing nontransfected cells. Cells were expanded and grown to bulk cultures in the presence of the selection antibiotic. EGFP-positive cells were subsequently enriched by sorting on a BD FACSAria flow cytometer. Unique stable cells are available upon request for noncommercial use.

All cells were maintained at 37 °C, 5% $CO_2$, and 3% $O_2$ in their specified growth medium.

**Patient-derived primary breast cancer cells**. Fresh scrapings were collected from primary breast cancer after surgical procedures in compliance with approved ethical permit. Permits were obtained from the regional ethics board at Karolinska Institutet in Stockholm and from the Stockholm medical biobank (ethical approval #02–061 with amendments 2013–1065–32 and 2015–2259; and ethical approval 2016/957–31 with amendment 2017/742–32). All patient material was anonymized. Single cells were generated immediately as previously described[48]. Cells were then differentiated in DMEM/F-12 GlutaMAX medium (10565, Life Technologies) supplemented with 5% FBS, 10 ng/mL epidermal growth factor (EGF, PHG0311, Life Technologies), 5 μg/mL insulin (I5500, Sigma), 1 μg/mL hydrocortisone (H0888, Sigma), and 10 μg/mL cholera toxin (C8052, Sigma). All the experiments were performed within five generations of these cultures that were then discarded.

**Murine-derived primary breast and pancreatic cancer cells**. The PyMT murine breast cancer cell line (PeRo-Bas1) was established from a FVB/N PyMT tumor[31]. Briefly, the tumor tissue was chopped in small fragments, suspended in DMEM (41965, Life Technologies) supplemented with 10% FBS and 100 U/mL penicillin-streptomycin (15140, Life Technologies) and filtered through a 100 μm cell strainer to remove bigger tissue parts. Cells were cultured overnight. Cell debris and contaminating immune cells or erythrocytes were removed through media change and phosphate-buffered saline (PBS) washes. Culture was continued until the cells reached 75% confluence, when contaminating fibroblasts were removed through brief trypsin treatment.

The KPC murine pancreatic cancer cell line[32] was grown in DMEM/F-12 (11320, Life Technologies) supplemented with 10% FBS and 100 U/mL penicillin–streptomycin (15140, Life Technologies).

**Xenograft model**. In total, $2 \times 10^6$ MCF7 cells stably expressing shControl ($n = 8$) or shPAK4 ($n = 10$) were suspended in growth factor-reduced Matrigel/PBS (100 μl, 50% final concentration, 356230, Becton Dickinson) and subcutaneously (s.c.) inoculated at the right flank of 6–8-week-old BALB/c female mice. Mice were implanted with 17β-estradiol pellets (0.72 mg per pellet, 90-day release, NE-121, Innovative Research of America) prior to the s.c. injection. After 9 days, when the cells started to form palpable tumors, xenograft size was determined regularly by externally measuring the growing tumors in two dimensions using an electronic caliper. Tumor volume ($V$) was approximated by the equation $V = \frac{1}{2} (L \times W^2)$, where $L$ is the length and $W$ is the width of the xenograft. All mouse experiments were done in a blinded fashion with mice being randomly selected for experiments. The person performing the measurements was blinded to the treatment groups. At the end of the experiment, xenografts were resected, photographed, weighted, and processed according to subsequent analyses.

**Intratumoral injection of PF-03758309**. PF-03758309 has been previously characterized as a PAK4 inhibitor[34]. PF-03758309 (Anthem Biosciences, Bangalore, India) was dissolved in dimethyl sulfoxide (DMSO, D5879, Sigma).

The compound was administered intratumorally at 6 mg/kg body weight in 100 μl of PBS to 10-week-old MMTV–PyMT females (FVB/N) with palpable tumors ($n = 8$). The largest palpable tumor at the experimental starting point was injected daily for 5 consecutive days. Vehicle animals ($n = 5$) received the same number of injections of the solvent. At the experimental endpoint, tumors were harvested and processed according to subsequent analyses.

**Small-interfering RNAs (siRNAs)**. The siRNAs used in this study were synthesized by Qiagen, Life Technologies and Genepharma. siRNA targeting sequences were as follows: PAK4#1: AACGAGGTGGTAATCATGAGG, PAK4#2: CGAGAA TGTGGTGGAGATGTA, RELB#1: UGCUGAACACCACUGAUAUGUCCUC, RELB#2: AAUUGGAGAUCAUC GACGAGU, CEBPB#1: CCGCCUGCCUUUAAAUCC, CEBPB#2: GGCCCUGA GUAAUCGCUU, PAK4m#1: CAGAGACGACACUAUGAGAAA, PAK4m#2: TT CGTGGGACCUACUACUGAA, Control: AAGCGCGCUUUGUAGGAUUCG or AllStars Negative Control siRNA (Qiagen, cat#SI03650318).

**sgRNAs**. For CRISPR/Cas9-mediated PAK4 gene ablation, two single guide (sg) RNA sequences against PAK4 (sgPAK4#1: 5′-TGATCGAGGAGTCGGCTCGC-3′ and sgPAK4#2: 5′-GTGCACGCGGTGCTCGAAGT-3) were designed with CRISPR DESIGN[49] (http://crispr.mit.edu). Plasmids for the lentivirus vector-mediated CRISPR-Cas9 (lentiCRISPRv2, cat#52961) and packaging (pCMV-VSV-G, cat#8454 and psPAX2, cat#12260, generously provided by Eyal Gottlieb, The Beatson Institute, Glasgow, UK). Oligonucleotides for sgRNA were annealed and cloned into the lentivirus transfer vector lentiCRISPRv2 at the BsmBI restriction site[50] and the specific target sequence was verified by DNA sequencing.

**Plasmids and mutagenesis**. FLAG–PAK4, FLAG–BAP, EGFP, EGFP–PAK4 wt, and EGFP–PAK4–M350 constructs have been previously described[51].

An siRNA-resistant form of PAK4 was generated by site-directed mutagenesis using the QuikChange Site-Directed Mutagenesis Kit (200518, Agilent) and the EGFP–PAK4 construct. The silent mutations were introduced into the siRNA

target region using the following primers: 5′-GCTCTTCAACGAGGTTGTTATCA TGAGGGAC-3′ and 5′-GTAGTCCCTCATGATAACAACCTCGTTGAAG-3′.

FLAG-tagged human RELB (FLAG-RELB-*wt*) was constructed by cloning the full-length human RELB cDNA[52] (gift from Sam Okret, Karolinska Institutet, Huddinge, Sweden) into the vector p3 × FLAG-CMV-10 (E4401, Sigma) within *Hind*III and *Kpn*I restriction sites to fuse them in N-terminal with FLAG. The primers used were 5′-TAGAAGCTTATGCT TCGGTCTGGGCCAGCC-3′ and 5′-GTAGGTACCCTACGTGGCTTCAGGCCC CGGGG-3′. The FLAG-tagged RELB mutants S151A and S151E (FLAG-RELB-S151A and FLAG-RELB-151E) were created by site-directed mutagenesis using the QuikChange Site-Directed Mutagenesis Kit (200518, Agilent). For S151A the following primers were used: 5′-GAGGGCCGCTCGGCCGGCGCCATCCTTGG GGGAGAGC-3′ and 5′-GCTCTCCCCAAGGATGGCGCCGGCCGAGCGGCCC TC-3′. For S151E the following primers were used: 5′-GAGGGCCGCTCGGCCGG CGAGATCCTTGGGGGAGAGC-3′ and 5′-GCTCTCCCCAAGGATCTCGCCGGC CGAGCGGCCCTC-3′.

GST-tagged murine RELB-N, -C, and -RHD (provided by Ralf Marienfeld, University of Würzburg, Würzburg, Germany) have been previously reported[53]. hRELB aa102–400 was PCR amplified from hRELB cDNA and cloned into the pGEX-4T-1 vector (28–9545–49, GE Healthcare) (named GST-RHD) within *Eco*RI and *Not*I restriction sites using the following primers: 5′-AAGGTAGAATTCGCC ACGCCGCCGCCTTGG-3′ and 5′-GAGTAGCGGCCGCCTAGCGAGGCAGGT ACGTGAAAGG-3′. GST-RHD-S151A was created by site-directed mutagenesis using the QuikChange Site-Directed Mutagenesis Kit (200518, Agilent) as above. The sequences of all primers used in this study can be found in Supplementary Table 5.

To construct His–PAK4 expression vector, full-length human PAK4 including a Strep II tag was amplified by PCR and cloned into pET15b (69661–3, EMD Millipore) within *Nde*I and *Bam*HI restriction sites to fuse it in N-terminal with 6 × His-tag.

All constructs have been sequenced.

**Transient transfection of siRNAs and plasmids.** Cancer cells were reverse transfected with siRNAs (10–30 nM) using Lipofectamine RNAiMAX transfection reagent (13778, Invitrogen) according to the manufacturer's instructions.

HMECs were transiently transfected with siRNAs using the HMEC Nucleofector Kit (VPK-1002, Lonza) according to the manufacturer's instruction. Program Y-001 was used for higher transfection efficiency.

Cells were transiently transfected with expression plasmids using Attractene transfection reagent (301005, Qiagen) or Lipofectamine 2000 or 3000 (11668 or L3000, Invitrogen) according to the manufacturer's suggestion. Routinely 8 µg of DNA was transfected for each 100 mm cell culture dish.

Cells were harvested for the various experimental purposes at the timepoints specified in the figures. The efficiency of gene suppression/expression was monitored after transfection by immunoblot, RT-qPCR, or immunostaining.

**CRISPR/Cas9 gene editing.** For *PAK4* gene ablation, we established BT-549 cells with stable expression of Cas9 and two single guide RNA sequences against PAK4 (sgPAK4). Specifically, lentiviral particles expressing sgPAK4 were generated in HEK293T cells by co-transfecting lentiCRISPRv2 vector, psPAX2, and pCMV-VSV-G at a ratio of 4:3:2 using Lipofectamine 2000 (11668, Invitrogen). After 24 h, the media was replaced with fresh media. The supernatant containing the viral particles was collected every 24 h for 2 days, filtered through a 0.45 µm pore membrane (83.1826, Sarstedt), and centrifuged 45 min at 1500 g at 4 °C with Lenti-X concentrator (631231, Clontech). The pellet, containing viral particles, was dissolved in medium and used for cell infection. Cells were transduced in 2 mL of media with 100 µl of viral supernatant in six-well plates. At 48 h post transduction, cells were selected with 1 µg/mL puromycin (P8833, Sigma) for 48 h, and then cells were expanded in the regular culture medium and used for assays at the timepoints specified in the figures. The effectiveness of CRISPR gene knockout was confirmed by immunoblotting with the indicated antibody.

**SA-β-gal activity.** For cultured cells, SA-β-gal staining was performed at the indicated time points using the Senescence β-Galactosidase Staining Kit (9860, Cell Signaling) or as described previously[29]. Images were acquired with a Nikon widefield, 10 × 0.45 NA air objective and MMI cell scan software. Regions of interest were selected according to a standardized pattern, in which X-Gal+ (5-bromo-4-chloro-3-indolyl-β-D-galactopyranoside) and the total number of cells were manually counted in ImageJ/Fiji using the Cell Counter plugin (National Institutes of Health, NIH) to determine the percentage of X-Gal+ cells.

An additional quantitative assay of SA-β-gal activity using cell extracts was employed as previously described[30]. This method uses 4-Methylumbelliferyl-β-D-galactopyranoside (MUG, M1633, Sigma) as substrate instead of X-Gal. The fluorescent product was measured using an automated plate reader (SpectraMax Gemini, Molecular Devices) with excitation at 360 nm and emission at 465 nm.

SA-β-gal staining of cryosections was performed on freshly harvested tissues at the experimental endpoint. Briefly, after resection, tumors were immediately snap frozen in liquid nitrogen, embedded in OCT cryomount (45830, Histolab), cryosectioned (4-µm thickness), and mounted onto SuperFrost Plus slides

(631–0108, VWR). Slides were immersed in fixation solution containing 2% formaldehyde (F1635, Sigma)/0.2% glutaraldehyde (G7776, Sigma) in PBS for 5 min at room temperature, washed with PBS, and stained in 40 mM citric acid/ sodium phosphate (pH = 5.5) containing 1 mM MgCl₂, 1 mg/mL X-Gal (B9146, Sigma), and 5 mM of each potassium ferricyanide (P-8131, Sigma) and potassium ferrocyanide (P-3289, Sigma) at 37 °C. Slides were counterstained with eosin (01650, Histolab) and imaged on a Zeiss Axiovert 100 inverted microscope. Images were deconvoluted with ImageJ/Fiji (National Institutes of Health, NIH) using the Color Deconvolution plugin[54]. The area occupied by X-Gal staining was thresholded and presented as a percentage of the total tissue area. The X-Gal+ area calculated per animal represents the average of two independent sections per tumor and the quantification of 2–10 nonoverlapping areas per section.

SA-β-gal staining of xenografted MCF7 tumors was performed on tissue slices of ~1-mm thickness, fixed in 2% formaldehyde/0.2% glutaraldehyde in PBS for 5 min at room temperature for immediate free-floating SA-β-gal whole-mount staining in the solutions described above. Tissues were photographed and the X-Gal+ area was determined as specified above for cryosections.

SA-β-gal staining of PyMT-driven early lesions was performed on whole mammary glands using the free floating SA-β-gal wholemount staining approach described above. X-Gal-stained mammary glands were then sectioned (4-µm thickness), counterstained with hematoxylin and imaged on Pannoramic MIDI II slide scanner from 3DHISTECH. Fifteen lesions per animal in close proximity to the nipple were selected and the number of X-Gal+ as well as the total number of cells per lesion were manually counted for determination of the percentage of X-Gal-positive cells per lesion.

**Immunostainings.** Xenografted tumor tissue was embedded in OCT (45830, Histolab), cryosectioned (10-µm thickness), and mounted onto SuperFrost Plus slides (631–0108, VWR). Sections were immersed in fixation solution containing 4% formaldehyde (F1635, Sigma) in PBS for 5 min at room temperature, washed with PBS and incubated with blocking solution consisting of 10% normal donkey serum (017–000–121, Jackson ImmunoResearch) in PBS with 0.3% Triton X-100 (X100, Sigma) for 1 h at room temperature. Sections were then incubated at 4 °C overnight in a humidified chamber with primary antibodies diluted in 10% normal donkey serum in PBS. Slides were probed with AF488-conjugated α-p53 (1:100, DO-1 #sc-126 AF488, Santa Cruz) and α-Ki67 (1:250, #ab15580, AbCam). After washing, antibody labeling was revealed using species-specific fluorophore-conjugated secondary antibodies (1:500, Rhodamine Red-X goat anti-rabbit IgG, 111–295–144, Jackson ImmunoResearch or Oregon Green 488 goat anti-Rabbit IgG, O-11038, ThermoFisher) in 10% donkey serum in PBS for 1 h at room temperature. Sections were counterstained with Hoechst 33342 (14533, Sigma). Images were acquired using a Nikon A1R confocal microscope with a Plan Apo VC 60×/1.4 NA oil objective and NIS-Elements software (Nikon).

Mammary tissues harvested from the transgenic mouse models were paraffin-embedded and sectioned at 4-µm thickness. For immunohistochemistry, sections were deparaffinized by incubation at 60 °C for 1 h followed by rehydration steps through washing in xylene and graded ethanol to distilled water. Samples were boiled for 20 min in antigen retrieval solution 0.01 M sodium citrate buffer (100813M, BDH) (pH 6.0) in water. Endogenous peroxidase activity was blocked via treatment with 3% hydrogen peroxide in water (H1009, Sigma). PBS diluted normal serum from the same host species as the secondary antibody was used as a blocking buffer. Slides were incubated at 4 °C overnight in a humidified chamber with a α-Cre antibody (1:100, PRB-106P, Covance) or newly generated α-PAK4 #73 antibody diluted 1:100 in blocking buffer. After three times PBS washing, slides were incubated at room temperature for 1 h with Biotin-SP-AffiniPure Goat α-Rabbit IgG (1:400, 111–065–144, Jackson ImmunoResearch) followed by three washes with PBS and then were incubated room temperature for 20 min with HRP-conjugated Streptavidin (016–030–084, Jackson). Following three washes with PBS, development was performed with diaminobenzidine (DAB, K3467, DAKO Sweden). Then slides were counterstained with Mayer HTX staining solutions (01820, Histolab), dehydrated, and mounted using DPX mounting media (44581, Sigma). Images were acquired on Pannoramic MIDI II slide scanner from 3DHISTECH. Scanned slides were further quantified using the HistoQuant module in Panoramic Viewer (3DHISTECH). H-Score was automatically calculated by the software using the following formula: H-Score = $(1 + i)Pi$ (where i is intensity score and Pi is the percentage of positivity).

**In situ hybridization (ISH) of PAK4 mRNA expression.** In situ hybridization for *PAK4* RNA transcripts was performed using the RNAscope 2.0 High Definition assay (Advanced Cell Diagnostics). Assays and quantifications were performed according to the manufacturer's instructions using a specific probe set against murine *PAK4* (Probe: *Mm-PAK4*). Positive and negative controls were *Mm-Polr2a* (312471) and *dapB* (310043), respectively. Slides were scanned using Pannoramic MIDI II from 3DHISTECH. Scanned slides were compared with positive and negative probes and were scored from very low expression (score 1) to highly expressed (score 3).

**Cell proliferation assay.** At the specified timepoints, cells growing on glass or plastic substrates, were cultured at 37 °C for 1 to 2 h in the presence of 30 µM 5-

ethynyl-20-deoxyuridine (EdU, Life Technologies) or 30 μM 5-Bromo-2'-deox-yuridine (BrdU, B5002, Sigma). Cells were fixed and assayed for EdU incorporation using an EdU Assay kit (C10635, Life Technologies) according to the manufacturer's instructions or cells were fixed and stained for BrdU using α-BrdU mAb G3G4 (1:100, G3G4, Developmental Studies Hybridoma Bank, Department of Biological Sciences, University of Iowa, Iowa City, IA) as described previously.

EdU/BrdU imaging was performed on a Nikon A1R confocal microscope using a Plan Apo γ 20X/0.75 NA air objective, producing a pixel resolution of 0.82 μm. In total, 3 × 3 or 4 × 4 image montages were acquired and stitched. Nuclear outlines were identified using Hoechst 33342 (14533, Sigma) staining. The percentage of EdU/BrdU$^+$ cells in total cells was analyzed with the NIS-Elements software (Nikon).

For EdU/BrdU analysis by flow cytometry a minimum of 10,000 events was analyzed on a BD FACSCalibur or a BD FACSCantoII.

HMECs harboring 4-Hydroxytamoxifen (OHT)-inducible RAS$^{G12V}$ were EdU-stained and analyzed by FACS 48 h after transient transfection with the indicated plasmids and simultaneously treated with 12.5 nM OHT (H7904, Sigma) or vehicle.

**Colony formation assay**. For colony formation assays on plastic six-well plates, cells were replated 4 days after lentiviral infection at single cell density (15,000 cells per well) in regular cell culture medium. At the indicated timepoints post-seeding, cells were rinsed twice with PBS, fixed with 4% formaldehyde (F1635, Sigma) for 20 min at room temperature and stained with 0.01% Crystal Violet in ddH$_2$O (0.01% [w/v], C3886, Sigma) for 30 min at room temperature. Plates were then washed extensively and allowed to dry. Plates were scanned and analyzed with ImageJ/Fiji software (National Institutes of Health, NIH) using the Colony Area plugin[55].

**Mammary gland wholemounts and sections**. The #4 inguinal mammary glands were dissected from 12 weeks old female mice ($n = 6$ per genotype), wholemounted onto SuperFrost Plus slides (631–0108, VWR), fixed in Carnoy's fixative (60% ethanol, 30% chloroform, and 10% glacial acetic acid) for 2 h at room temperature, stained with a solution of carmine (C1022, Sigma) and aluminum potassium sulfate (A7167, Sigma) overnight at room temperature, de-stained in 70% ethanol, further washed in 100% ethanol, cleared of fat in xylene (28975, VWR) overnight and mounted using DPX (44581, Sigma) or stored in xylene until scanning. ImageJ/Fiji (National Institutes of Health, NIH) was used for morphometric analysis. Raw images were cropped to include only the fourth mammary gland, converted to 8-bit and sharpened. The area occupied by carmine staining (mammary epithelium) was thresholded and presented as a percentage of the total mammary fat pad area.

Paraffin-embedded tissues (sectioned at 4-μm thickness) were routinely stained with hematoxylin and eosin for morphological evaluation according to standard procedures.

**Cell lysis**. Adherent or pelleted cells harvested by trypsinization (15400, ThermoFisher) were washed with PBS and lysed on ice in cold PAK-lysis buffer (50 mM Tris, pH 7.5, 150 mM NaCl, 5 mM MgCl$_2$, 1 mM EDTA, 10% glycerol, and 1% NP-40) or PBSTDS lysis buffer (1% Triton X-100, 0.5% sodium deoxycholate, 0.1% sodium dodecyl sulfate (SDS), 0.14 M NaCl, 2.8 mM KCl, 10 mM Na$_2$HPO$_4$, and 1.5 mM KH$_2$PO$_4$) containing freshly added cOmplete protease (1697498, Sigma) and phosphatase (P0044, Sigma) inhibitors cocktails.

Total protein from snap frozen mouse tissues was extracted with RIPA buffer (50 mM Tris, pH 7.5, 150 mM NaCl, 1% Triton X-100, 1% sodium deoxycolate, and 0.1% SDS) supplemented with protease and phosphatase inhibitor cocktails as above.

To prepare nuclear extracts, cells were lysed with cytoplasmic buffer (10 mM HEPES pH 7.5, 10 mM KCl, 1.5 mM MgCl$_2$ and freshly added protease and phosphatase inhibitors as above), incubated on ice for 15 min, homogenized in a glass Dounce homogenizer with 30 strokes of a tight-fitting glass pestle Kontes B, and checked under the microscope to verify that 95% of the cells displayed trypan blue (T8154, Sigma) staining. Nuclei were pelleted by centrifugation at 350 g for 5 min at 4 °C, washed with cytoplasmic buffer three times, dissolved in nuclear buffer (20 mM HEPES, 25% glycerol, 0.42 M NaCl, 1.5 mM MgCl$_2$, 0.2 mM EDTA, and freshly added protease and phosphatase inhibitors) and kept on ice for 30 min before vigorous vortexing. Nuclear extracts were then cleared at 16,000 g for 20 min at 4 °C.

Protein concentration was determined with the Pierce bicinchoninic acid assay (23225, ThermoFisher) kit according to the manufacturer's instructions.

Cell extracts were denatured by boiling at 95 °C upon addition of 5× SDS loading buffer (10% SDS, 20% glycerol, 0.05% bromophenol blue, 0.2 M Tris-HCl pH 6.8, and 10 mM β-mercaptoethanol). Otherwise, cells were lysed directly in 2× SDS loading buffer.

**Immunoblotting**. Equal amounts of denatured protein were subjected to 10% SDS-PAGE (SDS-polyacrylamide gel electrophoresis) using molecular weight markers (26619, Fermentas) to confirm the expected size of the target proteins and transferred to PVDF membranes (IPVH00010, Millipore). Membranes were washed in TBST buffer (TBS containing 0.1% Tween-20) and nonspecific binding sites were blocked by immersing the membranes in blocking buffer containing 4% nonfat

milk (70166, Sigma) in TBST buffer for 1 h on a shaker at room temperature or overnight at 4 °C. Membranes were probed with the following primary antibodies: α-PAK4 pab 6508 (1:1000), α-RELB (1:1000, C1E4 #4922, Cell Signaling; 1:1000, #06–1105, Millipore; 1:1000, EP613Y #GTX61291, Genetex and 1:1000, clone 17.3 #LS-C354950–100, LSBio), α-p53 (1:1000, DO-1 #sc-126, Santa Cruz); α-p21 (1:1000, C19 #sc-397, Santa Cruz); α-ACTIN (1:1000, JLA20, Developmental Studies Hybridoma Bank - DSHB); α-VINCULIN (1:100000, #V9131, Sigma); α-GAPDH (1:50000, #MAB374, Millipore) and α-pRb (1:1000, #554136, BD-Pharmingen). Antibodies against ACTIN, VINCULIN, GAPDH, and pRb served as control for protein loading. Next, appropriate peroxidase-conjugated anti-mouse (715–035–150, Jackson ImmunoResearch) or anti-rabbit (111–035–144, Jackson ImmunoResearch) secondary antibodies were used diluted 1:3000.

Bound antibodies were visualized with the Pierce enhanced chemiluminescence Plus Western Blotting Substrate detection system (32132, ThermoFisher) according to the manufacturer's instructions.

Quantity One analysis software (Bio-Rad) and ImageJ/Fiji (National Institutes of Health, NIH) were used for densitometric analysis of western blots. Quantification results were background subtracted and normalized to a loading control.

Uncropped and unprocessed scans of all blots are supplied in the Source Data file.

**Co-immunoprecipitations**. Immunoprecipitations were performed by incubating the same amount of the relevant whole-cell lysates with the appropriate antibodies (500–1000 μg of whole cell lysate, 2 μg of antibody) overnight at 4 °C with gentle agitation to allow immunocomplexes to form. Prior to incubation with the specified antibodies, samples were precleared by incubation with protein G-Sepharose beads (sc-2002, Santa Cruz) and 2 μg of α-mouse IgG (I5381, Sigma), for 1 h at 4 °C. Following the overnight incubation and when applicable, the immunocomplexes were collected by the addition of protein G-Sepharose beads for 3 h at 4 °C and washed six times with lysis buffer. Proteins were eluted by boiling in 2× SDS loading buffer and eluates were resolved on a 10% SDS-PAGE for immunoblotting with relevant antibodies. Unless otherwise indicated, FLAG-tagged proteins were immunoprecipitated with EZview Red ANTI-FLAG M2 Affinity Gel (F2426, Sigma). Otherwise, the following antibodies were used: monoclonal mouse α-PAK4 (clone OTI1C7 #CF807297, Origene), monoclonal mouse α-RELB (clone 17.3 #LS-C354950–100, LSBio), and α-mouse IgG (I5381, Sigma) as control.

**NF-κB DNA-binding activity**. DNA-binding activity of the mammalian NF-κB subunit RELB was measured in 20 μg of nuclear extracts using the TransAM NF-κB Family transcription factor assay kit (11467468, Active Motif) in accordance with the manufacturer's protocol.

**Exome sequencing**. Genomic DNA was extracted from flash-frozen liver tissue and from three mammary tumors that arose in virgin MMTV–PAK4 female mice (20–24 months old) using the DNeasy Blood & Tissue Kit (69504, Qiagen) as specified by the manufacturer. DNA concentrations were determined using an Infinite®200 PRO (Tecan) plate reader. SOLiD Fragment Standard Library Builder and SureSelect Mouse Exome kit (Agilent) were used according to the manufacturer's protocol to prepare samples for WES performed on a SOLiD 5500 machine at Science for Life Laboratories, Uppsala, Sweden, producing unpaired, matched reads.

Exome sequencing reads were aligned to the mouse mm9 genome using BWA (bio-bwa.sourceforge.net/) with additional processing using Picard (broadinstitute.github.io/picard/) and Samtools[56]. Somatic variants were called using MuTect[57] and annotated using Annovar[58].

**RNA sequencing**. Total RNA was isolated from Hs 578T and BT-549 cells 72 h after transient transfection with PAK4-targeting (siPAK4#1) or control siRNA using the RNeasy Mini Kit (74104, Qiagen) as specified by the manufacturer. RNA quality was assessed with Agilent 2100 Bioanalyzer (Agilent Technologies) prior to high-throughput mRNA-sequencing of poly-adenylated transcripts from total RNA using paired end sequencing on an Illumina HiSeq2000 platform. Library construction and sequencing was performed at the SciLifeLab in Stockholm.

Paired end reads were aligned to the human genome (hg19) using Tophat2[59] retaining only uniquely mapping reads. Reference splice junctions were provided via reference transcriptome (Ensembl build 73). Explicit read abundance at each transcript model was assessed using HTSeq (python) prior to calling differential expression with DESeq2[60]. Significance thresholds were considered using an FDR corrected $p$-value ≤ 0.05. FPKM values (fragments per kilobase of exon model per million mapped fragments) were calculated using Cufflinks[61]. Figures were generated using the ggplot2 package in R. Gene ontologies were analyzed through the use of IPA[27] (QIAGEN Inc).

The list of NF-κB target genes was derived from a third party dataset (in fetal lung fibroblasts—https://www.ncbi.nlm.nih.gov/geo/query/acc.cgi?acc=GSM1055810). Here, raw sequences were aligned against the hg19 human genome using bowtie2 and peaks called using MACS (version 1.4) on standard settings. Our list of NF-κB targets was acquired by intersecting the known coding

genes in Ensembl mouse build 73 with our NF-κB subunit (p65) peaks using bedtools (intersect).

The replicate BT-549 siControl #3 was incorrectly sequenced and was therefore removed from the analyses.

**Reverse transcription quantitative PCR array (RT-qPCR)**. For RNA-Seq validation, a fast 48+ TaqMan-based qPCR array (4413257, Life Technologies) was customized and used to profile Hs 578T cells upon siRNA-mediated PAK4 depletion. This array was later employed to profile Hs 578T cells upon transient transfection of FLAG-RELB-wt versus FLAG-RELB-S151E.

The array consisted of a selection of 44 genes that were found DE upon PAK4 knockdown by RNA-Seq and that have been shown to play a role in senescence-associated phenotypes plus four housekeeping genes. The following TaqMan gene expression assays were used: 18s rRNA (Assay ID: Hs99999901_s1, housekeeping gene), BID (Assay ID: Hs00609632_m1), BIRC3 (Assay ID: Hs00985031_g1), BRAT1 (Assay ID: Hs00378008_m1), CCNG1 (Assay ID: Hs00171112_m1), CD82 (Assay ID: Hs01017982_m1), CDK1 (Assay ID: Hs00938777_m1), CDKN1C (Assay ID: Hs00175938_m1), CDKN2D (Assay ID: Hs00176481_m1), CEBPB (Assay ID: Hs00270923_s1), CSF1 (Assay ID: Hs00174164_m1), CYTL1 (Assay ID: Hs01573280_m1), DSCC1 (Assay ID: Hs00225430_m1), E2F1 (Assay ID: Hs00153451_m1), E2F2 (Assay ID: Hs00231667_m1), E2F3 (Assay ID: Hs00605457_m1), E2F6 (Assay ID: Hs00242501_m1), FANCM (Assay ID: Hs00326216_m1), GAPDH (Assay ID: Hs99999905_m1, housekeeping gene), GUSB (Assay ID: Hs99999908_m1, housekeeping gene), HES1 (Assay ID: Hs00172878_m1), HEY1 (Assay ID: Hs01114113_m1), HEY2 (Assay ID: Hs00232622_m1), HPRT (Assay ID: Hs99999909_m1, housekeeping gene), ICAM1 (Assay ID: Hs00164932_m1), IL8 (Assay ID: Hs00174103_m1), LIG1 (Assay ID: Hs01553527_m1), MAP2K6 (Assay ID: Hs00992389_m1), MCM10 (Assay ID: Hs00960349_m1), NFKB1 (Assay ID: Hs00765730_m1), NFKB2 (Assay ID: Hs01028901_g1), OSGIN1 (Assay ID: Hs00203539_m1), PAK4 (Assay ID: Hs01100061_m1), PCNA (Assay ID: Hs00427214_g1), POLA1 (Assay ID: Hs00415835_m1), PRKDC (Assay ID: Hs00179161_m1), RBL1 (Assay ID: Hs00765700_m1), RELB (Assay ID: Hs00232399_m1), S100A4 (Assay ID: Hs00243202_m1), SNAI2 (Assay ID: Hs00950344_m1), SPP1 (Assay ID: Hs00959010_m1), TIMP2 (Assay ID: Hs00234278_m1), TIPIN (Assay ID: Hs00762756_s1), TNFRSF11B (Assay ID: Hs00900358_m1), TONSL (Assay ID: Hs00273774_m1), TP53 (Assay ID: Hs01034249_m1), TRAF2 (Assay ID: Hs00184192_m1), and TWIST1 (Assay ID: Hs01675818_s1).

Total RNA was isolated using the RNeasy Mini Kit (74104, Qiagen) as specified by the manufacturer. RNA quality was assessed with Agilent 2100 Bioanalyzer (Agilent Technologies) prior to the qPCR array run according to the manufacturer's instructions at the BEA core facility in Stockholm. RT-qPCR reactions were carried out in biological triplicates.

**Recombinant proteins and protein purification**. GST-tagged human RELB (GST-RELB) was commercially available (H00005971-P01, Tebu-bio). GST and GST-tagged murine RELB-N, -C, and -RHD (GST-RELB-N, GST-RELB-C, and GST-RELB-RHD)[53] were purified with Glutathione Sepharose 4B (17–0756–01, GE Healthcare) according to the manufacturer's instructions. Briefly, plasmids were transformed into BL21 Star (DE3) *Escherichia coli* (C601003, ThermoFisher) and the bacteria were grown in Luria–Bertani broth plates and media in the presence of ampicillin (A9518, Sigma). Protein expression was induced in early log phase cultures with 0.1 mM IPTG (I5502, Sigma). After 4 h at 37 °C, cells were harvested and lysed by sonication in ice-cold PBS containing cOmplete protease inhibitor cocktail (1697498, Sigma). Triton X-100 (X100, Sigma) was added to a final concentration of 1% and GST-tagged proteins were purified with Glutathione Sepharose 4B (17–0756–01, GE Healthcare) according to the manufacturer's instructions. Expression of GST-tagged hRELB aa 102–400 (GST-RHD) and mutant GST-RHD-S151A was induced with 0.2 mM IPTG (I5502, Sigma) at 16 °C overnight.

Recombinant His-tagged PAK4 was produced by expressing the His-PAK4 construct in BL21Star (DE3) *Escherichia coli* as described earlier[51]. Plasmid expression was induced by 0.5 mM IPTG and the bacteria were grown for 4 h at 37 °C prior to lysis with 0.3 M NaCl, 50 mM Tris-HCl pH 8.0 and sonication on ice. The protein was then purified on an HIS-Select HF Nickel Affinity Gel (H0537, Sigma) in accordance with the manufacturer's protocol. Purified protein was dialyzed against PBS and concentrated.

**Generation of anti-PAK4 antibodies**. For anti-PAK4 antibody production, a PAK4 $NH_2$-terminal sequence (aa 1–326) was amplified by PCR and cloned into the PET15b vector (Novagen). His-tagged fusion protein was expressed in BL21 De3Lys strain of *Esherichia coli*, solubilized in 50 mM Tris-HCl pH 8.0 with 0.3 M NaCl buffer and purified by absorption-elution on HIS-select HF Nickel affinity gel (Sigma) followed by dialysis against PBS. This PAK4 fusion protein in its native state has been used to generate anti-PAK4 6508. This fusion protein was fixed with PFA prior to injection into rabbits to generate anti-PAK4 serum #73. The IgG fraction from serum was purified on an affinity column consisting of the full-length PAK4 protein (prepared as above) coupled to CNBr-activated Sepharose 4B

(Amersham), generating anti-PAK4 pabs 6508 and 73. Sera are available upon request for noncommercial use.

**In vitro kinase assays**. In vitro radioactive kinase assays were performed where purified His-tagged PAK4 was incubated with GST-tagged RELB proteins as putative substrates. In brief, 2 μg of the indicated GST fusion proteins were incubated with 2 μg of purified His-tagged PAK4 in PAK-kinase buffer (100 mM HEPES pH7.5, 20 mM $MgCl_2$, 4 mM $MnCl_2$, and 0.4 mM DTT). Kinase reactions were performed in the presence of 5 μCi [γ-$^{32}$P]-ATP (NEG502A250UC, Perki-nElmer) and 60 μM cold ATP (A3377, Sigma) at 30 °C for 30 min. The reactions were stopped by boiling at 95 °C in 5× SDS loading buffer. Samples were resolved by SDS-PAGE; the gels were fixed in 7% acetic acid and 20% methanol and stained with Coomassie Blue dye (443283M, VWR). The radioactivity incorporated into the substrate was visualized and quantified by autoradiography and Phosphor-Imager analysis (Molecular Imager FX, Bio-Rad).

**Mass spectrometric analysis**. GST-RHD (GST-tagged hRELB aa 102–400) was phosphorylated by His–PAK4 in vitro and analyzed by MS.

Gel lanes with proteins were excised manually from Coomassie-stained gels. The lanes, cut in several pieces, were processed and digested by trypsin using a robotic protein handling system (MassPREP, Waters). Proteins were reduced with 10 mM dithiothreitol (DTT, Sigma) in 100 mM ammonium bicarbonate and incubated at 40 °C for 30 min 55 mM iodoacetamide (Sigma) in 100 mM ammonium bicarbonate was added for alkylation at 40 °C for 20 min Then 0.3 μg of sequencing grade modified trypsin (Promega) was added to each gel piece and incubated for 4.5 h at 40 °C. Peptides were extracted with 30 μl of 5% formic acid/ 2% acetonitrile, followed by extraction with 24 μl of 2.5% formic acid/50% acetonitrile, cleaned with C18 StageTip and dried the peptide extracts using a Speedvac before resuspending in 0.1% Formic Acid and 2% acetonitrile.

Chromatographic separation of peptides was achieved using 50 cm EASY-spray C18 column connected to EASY-nLC 1000 system (ThermoFisher). Eluted at 300 nL/min flow rate for 60 min at a linear gradient from 2% to 26% ACN in 0.1% formic acid. Orbitrap Fusion mass spectrometer (ThermoFisher) analyzed the eluted peptides that were ionized with electrospray ionization. The survey MS spectrum was acquired at the resolution of 120,000 in the range of m/z 350–1500. MS/MS data were obtained with a higher-energy collisional dissociation (HCD) and electron-transfer dissociation (ETD) for ions with charge z > 1 at a resolution of 30,000.

The raw data files were converted to Mascot Generic Format (mgf) using in-house written Raw2mgf program. Proteins were identified by searching mgf files w against SwissProt database and with variable modifications, Deamidated (NQ), Oxidation (M), Phospho (ST), Phospho (Y), using Mascot v 2.3.0 (MatrixScience) database search engine.

**Detection of phosphorylation sites within RELB**. The *RELB* gene was searched for in the Phosida[38] and PhosphoSitePlus[62] databases in accordance with the suppliers' guide. Literature was curated for additional residues. Residues previously identified in murine RELB were converted to the corresponding human residue for display.

**Clinical data and gene expression profiling**. The role of *PAK4* gene expression in clinical samples was explored on the previously published METABRIC data[14]. Briefly, 1980 patients were included in this study that performed transcriptional profiling (Illumina HT-12 v3 platform) of 1992 primary tumor specimen and tumor adjacent normal tissue (144 samples) from biobanks in the UK and Canada.

All clinical-pathological characteristics, PAM50 subtypes, Integrative subtypes, and survival outcome information were obtained from the original publication.

The normalized data were downloaded from the European genome-phenome archive platform (https://www.ebi.ac.uk/ega/studies/EGAS00000000083, accessed on Mars 2015). The 3 PAK4 annotated probes had Spearman correlation coefficients 0.55, 0.03, and −0.03. The most varying probe that targets all PAK4 transcript variants (ILMN_1728887) was selected for further analysis. A gene expression based proliferation module score was computed as previously described[35].

*PAK4* gene expression was explored in the Oncomine cancer microarray database and web-based data-mining platform (accessed in December 2016)[63]. PAK4 mRNA levels were compared in breast carcinomas versus normal breast tissues in the TCGA[16] (n = 93 carcinomas versus n = 61 normal) and Zhao[17] (n = 38 carcinomas versus n = 3 normal) datasets.

The somatic mutation and copy number segmentation data of the breast cancer cell lines were downloaded from the CCLE data portal (www.portals.broadinstitute.org/ccle)[64]. Prior knowledge about and recurrence of mutations were analyzed using OncoKB[65] (www.oncokb.org), Cancer Hotspots[66,67] (www.cancerhotspots.org), and 3D Hotspots[68] (www.3dhotspots.org). Copy number data were post-processed by using the GISTIC2.0[69] module on GenePattern[70] (www.genepattern.broadinstitute.org) running default parameters. Copy number alteration and clinical data of the TCGA breast cancer samples (n = 987) were downloaded from Broad GDAC Firehose (https://ezid.cdlib.org/id/doi:10.7908/C11G0KM9) (www.gdac.broadinstitute.org). Patients were stratified according to their PAK4

copy-number status (all other versus amplification) and further analyzed using the Kaplan-Meier method and compared by the logrank (Mantel-Cox) test carried out with SPSS Statistics software version 23 (IBM Corporation, Armonk, NY, USA).

**Statistics**. Group size was based on previous experience. No statistical method was used to predetermine sample size. Unless otherwise noted, each experiment was repeated three or more times. Data shown in column graphs represent mean ± standard error of the mean (SEM) or mean ± standard deviation (SD), as indicated in the figure legends, and individual data points are plotted. Statistical analysis was performed with GraphPad Prism 6.0. Details of statistical testing can be found in the figure legends and in the source data file. All datasets were tested for Gaussian distribution using the Kolmogorov–Smirnov test.

Statistical analysis of the METABRIC clinical data is described in detail below. The association between PAK4 and clinical-pathological characteristics was assessed by Mann–Whitney tests (two groups) or Spearman correlation test (continuous variable). OS and DSS endpoints were used to explore the prognostic ability of PAK4. Deaths of an unknown cause were excluded in analyses of the DSS endpoint. Some sensitivity analyses including deaths of an unknown cause as an event of the DSS endpoint demonstrated consistency in the results. In all patients as well as in the subgroup of untreated patients (not receiving any adjuvant treatment), PAK4 association with both endpoints was investigated in univariable and multivariable Cox models stratified by site (as suggested in the original publication). The multivariable Cox models were adjusted by tumor size, lymph node status, tumor grade, age at diagnosis, and PAM50 subtype. The univariable prognostic role of PAK4 categorized according to quartiles was visualized by the Kaplan–Meier method.

Association between PAK4 expression and modules that reflect biological processes in breast cancer[35,36], NF-κB signaling[37], and NF-κB subunits in tumor clinical samples was quantified by Spearman's rank correlation.

The analysis is exploratory. All statistical analyses were done with R statistical software v3.1.0.

**Reporting summary**. Further information on research design is available in the Nature Research Reporting Summary linked to this article.

## Data availability
The RNA-Seq data reported here has been deposited in the NCBI Geo under ID code: GSE112817. Exome-Seq data are available in SRA under the accession code PRJNA545882. Remaining primary data of interest is provided in the source data file of this paper.

## Code availability
R code used in this study has been deposited on GitHub (https://github.com/CostaNatCommun19/analysisPublicData).

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

## Acknowledgements

We thank Audrey Minden, Lars Holmgren, Rainer Heuchel, Matthias Löhr, David Beach, Sam Okret, and Ralf Marienfeld for providing various mice, cells, and reagents as specified in Methods. We thank Zhilun Li for assistance with kinase assays, Agneta Andersson for support with the xenografts and for performing ISH, Isha Raj for help with the structural modeling, Sara Göransson and Xavier Serra-Picamal for assistance with imaging and image analysis of X-Gal experiments in cells, and Jens Henrik Norum and Erik Fredlund for assistance with exome sequencing analysis. We thank all Strömblad laboratory members for valuable comments and discussion. This study was supported by grants to S.S. from the Swedish Research Council, Radiumhemmets Forskningsfonder and the Swedish Cancer Society and the Breast Cancer Theme Center at KI. T.C. was supported by the Portuguese Foundation for Science and Technology (SFRH/BD/47330/2008). The mAbs anti-ACTIN JLA20 and anti-BrdU G3G4 were obtained from the Developmental Studies Hybridoma Bank, created by the NICHD of the NIH and maintained at the University of Iowa, Department of Biology, Iowa City, IA 52242. This study makes use of data generated by the Molecular Taxonomy of Breast Cancer International Consortium supported by Cancer Research UK and the British Columbia Cancer Agency Branch. Microscopy was performed at the Live Cell Imaging facility/Nikon Center of Excellence, Department of Biosciences and Nutrition, Karolinska Institutet, Huddinge, Sweden, supported by grants from the Knut and Alice Wallenberg Foundation, the Swedish Research Council, the Centre for Innovative Medicine, the board of research at KI and the Jonasson donation to the School of Technology and Health, Royal Institute of Technology, Sweden. We also thank the core facility Bioinformatics and Expression Analysis, which is supported by the board of research at the KI and the research committee at the Karolinska Hospital. Proteomic analysis was carried out at the Proteomics Karolinska. The authors also acknowledge support from Science for Life Laboratory, the National Genomics Infrastructure, NGI, and Uppmax for providing assistance in massive parallel sequencing and computational infrastructure. Open access funding provided by Karolinska Institute.

## Author contributions

T.C. conceived, designed, and performed most experiments and analyses. T.Z., H.O., M.M.B., M.Z., P.R., R.K., M.S., P.H.V., U.R. and X.G. contributed to experiments and analyses. J.L. and J.S. analyzed clinical data. E.T. and P.D. contributed the MMTV–PAK4 transgenic mouse model. N.R. performed RNA-Seq analyses. R.M. and J.H. provided PDCs. P.R. and K.P. provided PyMT-derived samples and cells. P.D.A. advised and assisted in data interpretation. S.S. conceived, designed and supervised the study, interpreted data, and provided financing. T.C. and S.S. wrote the manuscript and all authors approved the final version.

## Additional information

**Competing interests:** The authors declare no competing interests.

