## [Peer Review File · Nature Communications]

Reviewers' comments:

Reviewer #1, Expertise: PAX biology, cancer (Remarks to the Author):

This work describes the role of the group B Pak, PAK4, in breast cancer. It relies mainly on in vivo mouse models, supplemented by cell line work to get at molecular mechanisms. In general, I think they do a good job in presenting a plausible model of PAK4 function in breast cancer, but there are some missing controls and missed opportunities that should be addressed.

Major:

1) Missing control: The main missing element in this work is the failure to distinguish potential catalytic vs non-catalytic functions for PAK4. The Paks generally have two functions, as enzymes and as scaffolds, and little is done in this work to work out which is operative in the phenotypes of interest. While it might be too much to ask for the authors to make a second transgenic mouse expressing kinase-dead PAK4 (though it would be quite interesting), they certainly can do this experiment in cell culture models (e.g., in Fig 3). I don't think the experiments with the Pak inhibitor PF03758309 (Fig 4) are probative, as this compound is extremely dirty and the results they obtained could be due to inhibition of any of dozens of kinases.

2) Missed opportunity: given that the authors analyzed a PAK4-deleted GEM model in Fig. 2, why don't they examine the tumors that form for evidence of proliferation and senescence, as they for the xenografted mice in Fig. 4?

Minor:

1) The effects on promoting breast tumors in PAK4 transgenic are weak, on par with previous reports of PAK1. The latter paper (Kumar's group, early 2000s) should be cited and discussed briefly either in the Intro or Discussion section.

2) Similarly, PAK4 is often overexpressed in certain tumors due to gene amplification. I gather that is not the case in these breast cancers (whereas the PAK1 gene is often amplified in this setting), but this issue should be explicitly presented so the reader has context about the relative roles of Group A and B PAKs in breast cancer.

3) In Figs S2H and S2I, what is the explanation for the variation in PAK4 expression. Incomplete deletion of the floxed gene by CRE?

Reviewer #2, Expertise: senescence (Remarks to the Author):

This manuscript by Costa et al. is about p21-activated kinase (PAK) 4, which is known to be overexpressed in various types of cancer, including breast cancer. PAKs have been linked to cell proliferation and metastasis, as well as drug resistance in cancer.

From database analysis, the authors confirmed that PAK4 is upregulated in breast tumor patient samples and cell lines. Higher PAK4 expression level also correlated with worse prognosis. They generated two mouse models with altered PAK4 level: one is mammary-specific PAK4-transgenic mouse that developed mammary tumor with 25% penetrance, and another is PyMT-driven breast cancer mouse model (activates Ras and PI3K) combined with conditional PAK4 knock-out, in which tumorigenesis was slower and the survival time was extended. When PAK4 was knocked-down in cancer cell lines, senescence-like features were observed and cell proliferation was inhibited showing the anti-tumor growth effect of PAK4 inactivation.

PAK4 knock down correlated with NF- κ B target gene activation, and patient sample analysis revealed the reverse correlation between PAK4 expression and NF- κ B signature. Furthermore, they showed that PAK4 physically interacts with RELB, phosphorylate at a specific site, which in turn,

decreases RELB–DNA binding and target gene transcription, including CEBP β .

Overall, this is a thorough study about PAK4, encompassing patient sample analysis, mouse model, as well as biochemical experiments to reveal novel mechanism of PAK4 action. Particularly, the link between PAK4 and NF- κ B signaling which have been correlated, but not specifically shown, is now revealed in a very convincing manner.

However, some aspects need to be further clarified before this manuscript can be considered for the publication. Most of them are related to the expression level and activation of PAK4 and RELB, and PAK4's role in senescence. In addition, the analysis method of RNS sequencing data should be better clarified. See below.

Major concerns

1. Throughout the manuscript, I was wondering about the connection between the protein level and activation of PAK4. In patient samples, it is the increased expression that is relevant and in most of the experiments, only PAK4 expression level was modified and monitored. In the mechanical study in NF- κ B signaling, it is also not clear about the activation status of PAK4. Is activation by Ras not relevant for PAK4's role in cell proliferation and senescence? And how important is its role in Ras-induced senescence?
2. (related to 1.) Breast cancer cell lines have high PAK4 level. Do they also have Ras/Raf activation mutation? It looks like, only high level does not always cause tumor (see mouse model low penetrance). Therefore, it must require additional activation probably by Ras. Has it been tested? For example, by PAK4 overexpression in PyMT model? Or database analysis?
3. In the PAK4 overexpression transgenic model, some of the mice developed spontaneous oncogenic K-Ras mutation, which is constitutively active, and in combined with increased PAK4 level, it would lead to more activated PAK4. Was there any consequence in tumor biology due to activated PAK4?
4. Figure S2H: According to this result, some of PyMT;PAK4MEp^{-/-} mice still retain PAK4 expression. Obviously, this is an issue when interpreting the Kaplan-Meier plot (Fig. 2F-H) as well as histological analysis (Fig. 2D and E). Have the authors consider stratifying the result in terms of PAK4 level in tumors? Was there any correlation between PAK4 level in tumor and the tumor onset or survival time?
5. MMTV-PyMT is an aggressive metastatic breast cancer model. Considering the known function of PAK4 in cell proliferation and migration, it is plausible to expect that PyMT;PAK4MEp^{-/-} mice would have less metastasis. Has this been addressed? Also, in relation with the PAK4 level (see comment above)?
6. Figure 3C and S3C: how was the 'senescence signature' defined? Just 'literature review' doesn't seem to be as strong and objective basis. To be clear, the direction of gene regulation – what is up or down in senescence – should be annotated and compared as two separate signatures. Is there overlap of senescence state and the genes upregulated in PAK4 knock down cells? Moreover, how much two cell lines Hs578T and BT-549 share this signature in the same direction is not mentioned. I am also not convinced about this strongly separating heatmap, considering only part of the population entered senescence after almost complete knock down of PAK4 (e.g. ca. 50% cells are still BrdU-positive in all results, only a few cells are SA-b-gal positive in Fig S3E).
7. Figures S3E – N: the result of 4-MU staining, implying senescence, is shown only fold induction compared to the control. To see the extent of senescence reaction, it would be more helpful to show how many cells in certain population is actually senescent. Therefore, I would recommend

SA- β -gal staining or other method to show individual cells in or not in senescence.

8. Figure 3G-J: again, the impact of siPAK4 is only partial. To show that this is actually working, it would be important to show that siPAK4 cells, and only those with true knock-down proliferate in presence of RAS. In addition, siPAK4 seems to have pro-proliferatory effect on the epithelial cells (J), therefore, it is also possible to postulate that the result is only the sum of increased (by siPAK4) and decreased (Ras) proliferation of the whole cell population, not by direct interaction of those two moieties. Cell sorting for double positive cells, and also for formal proof, PAK4 activation after Ras induction would be recommended.

9. How fast is PAK4 in action? Cells with PAK4 CRSPR-CAS9 knock-down (Fig 3H) and xenograft of shPAK4 cells actually grow to a certain extent. Particularly for Xenograft, it seems they increase the tumor volume initially, although one assumes that the PAK4 was already inactivated at the time of inoculation.

10. Fig 5A and S5A: According to the legend, this is a clustered heatmap of 'NF-kB target genes that are differentially expressed', of which some are upregulated upon siPAK4. However, about 50% of the genes shown here are downregulated. What are those, NF-kB targets but regulated in significantly opposite direction upon PAK4 knock-down?

11. Related to 10, the authors describe in the methods section that the list of NF-kB target genes was derived from the public dataset (which is human sequences) and then the 'raw sequences (RNA-seq data from HS578T?) were aligned against the mm9 mouse genome...', which confused me further. Why did they use mouse genome? This has to be better clarified for readers who may not be an expert in RNA expression data analysis.

12. Fig 5E and S5E: The authors claim that PAK4 negatively regulate NF-kB and senescence-like arrest. Therefore, when there is less PAK4 (siPAK4), there are more NF-kB expressed, more p53, cells proliferate less (senescence?). Does this mean, more PAK4 leads to less Ras-induced senescence? But Ras activates PAK4. How can one solve this discrepancy?

13. It seems that PAK4 regulates RELB transcription (RNA-seq result and RT-qPCR). However, the mechanism suggested in Fig 6 is about RELB inactivation by PAK4. Please clarify this difference.

Minor comments

1. In the PyMT-induced breast tumor, what was the reason of PAK4 induction? Was amplification of the locus as found in patient samples?

2. Figures 2F, G, and H: the number of mice shown here should be indicated.

3. Figure 2H: what is the meaning of this result? What kind of value does it add to the story?

4. Line 201 – 205: 'Five breast cancer cell lines..... metastatic potential'. It would be helpful if this statement is supported by a supplementary figure or table. In addition, Ras/Raf mutational status should be also listed, since this is important for PAK4 activation. For their senescence potential, and particularly related to in vivo analysis, p53 mutational status should be also indicated for breast cancer cell lines.

5. Line 205: Figure S3D is not a SA- β -gal staining result.

6. Figure s3E: please quantify the result.

7. Figure 3D-F: senescence is a terminal cell cycle arrest and it is important to show that cells are

indeed arrested in G1. Please do cell cycle analysis of siPAK4 cells.

8. Figure 3G-H: similar to the comment above, please show G1 cell cycle arrest and SA-b-gal staining result to prove that this is senescence(-like) arrest.

9. Figure 4C: please indicate the sample number and also quantify the result.

Response to comments from the reviewers

Reviewers' comments:

Reviewer #1, Expertise: PAX biology, cancer (Remarks to the Author):

This work describes the role of the group B Pak, PAK4, in breast cancer. It relies mainly on in vivo mouse models, supplemented by cell line work to get at molecular mechanisms. In general, I think they do a good job in presenting a plausible model of PAK4 function in breast cancer, but there are some missing controls and missed opportunities that should be addressed.

We were happy that the reviewer is pleased with our model of PAK4 function in breast cancer. We are grateful for the comments as they helped us to clarify the raised points, which improved the manuscript.

Major:

1) Missing control: The main missing element in this work is the failure to distinguish potential catalytic vs non-catalytic functions for PAK4. The Paks generally have two functions, as enzymes and as scaffolds, and little is done in this work to work out which is operative in the phenotypes of interest. While it might be too much to ask for the authors to make a second transgenic mouse expressing kinase-dead PAK4 (though it would be quite interesting), they certainly can do this experiment in cell culture models (e.g., in Fig 3). I don't think the experiments with the Pak inhibitor PF03758309 (Fig 4) are probative, as this compound is extremely dirty and the results they obtained could be due to inhibition of any of dozens of kinases.

We thank the reviewer for this relevant question. In order to distinguish potential catalytic *versus* non-catalytic functions for PAK4 in the senescence-like growth arrest phenotype herein studied, we transiently overexpressed a kinase-dead EGFP-PAK4 M350 mutant in MCF7 breast cancer cells. Cells expressing EGFP and EGFP-PAK4 *wt* were used as controls. Our analysis indicate that the proliferation of MCF7 breast cancer cells expressing high EGFP-PAK4 M350 is impaired to an extent similar to that of siPAK4-transfected cells (in Fig. 3l and Supplementary Fig. S3h). This suggests that PAK-M350 acts as a dominant negative with respect to the studied phenotype and thus that the PAK4 kinase activity is relevant for the proliferation block.

2) Missed opportunity: given that the authors analyzed a PAK4-deleted GEM model in Fig. 2, why don't they examine the tumors that form for evidence of proliferation and senescence, as they for the xenografted mice in Fig. 4?

We agree that this is an interesting opportunity. When considering this opportunity, we need to keep in mind that there are major differences between these two models with respect to the degree of synchronicity of the tumor growth and PAK4-inhibition between different mice. While tumor growth as well as PAK4-inhibition is expected to be fairly synchronous in the xenograft model, the endogenously developed PyMT-driven tumors appear at different time points in different mice and in different glands of the same mouse. Also, each gland contains multiple tumors that differ in stage and size and the tumors themselves are heterogeneous. In addition, the

MMTV-Cre penetrance is heterogeneous in this model (discussed in minor point #3 below), resulting in somewhat heterogeneous PAK4 gene depletion as well. These different heterogeneities limit our expectations in terms of how clear differences we can expect to detect when comparing glands from different mice.

Nevertheless, as suggested by the reviewer, we have now analyzed mammary tissues harvested at 8-11 weeks of age from our MMTV-PyMT model with conditional PAK4 depletion, using X-Gal as a marker of senescence. The time point was selected due to previous evidence that senescent cells are abundant in the pre-neoplastic stage of certain cancers (such as in nevus that precede melanoma) (Michaloglou *et al.*, Nature 2005). While most early lesions displayed X-Gal positivity, we found that there is an increase, albeit modest, in X-Gal positive cells within lesions in PyMT;PAK4^{MEP^{-/-}} glands as compared to control PyMT; PAK4^{MEP^{+/+}} mice (Fig. 4a-c).

Minor:

1) The effects on promoting breast tumors in PAK4 transgenic are weak, on par with previous reports of PAK1. The latter paper (Kumar's group, early 2000s) should be cited and discussed briefly either in the Intro or Discussion section.

We thank the reviewer for pointing this out. Previous work from Kumar's group show that overexpression of a catalytically active mutant PAK1 leads to the development of malignant mammary tumors in 20% of transgenic females (Wang *et al.*, Oncogene 2006). Thus, the relatively low penetrance and long latency phenotype of their mouse model overexpressing catalytically active PAK1 is comparable to the phenotype of our transgenic model overexpressing *wt* PAK4. This suggests that in both cases, additional genetic mutations are likely needed for tumorigenesis. As suggested, we have now cited this work and briefly discussed it in the Discussion section of the revised manuscript.

2) Similarly, PAK4 is often overexpressed in certain tumors due to gene amplification. I gather that is not the case in these breast cancers (whereas the PAK1 gene is often amplified in this setting), but this issue should be explicitly presented so the reader has context about the relative roles of Group A and B PAKs in breast cancer.

We agree that this is of interest and have added analysis of PAK4 amplification in two large breast cancer patient datasets.

PAKs overexpression in cancer varies widely and may be due to both mRNA upregulation and/or gene amplification (Kumar & Li, Advances in Cancer Research 2016). PAK1 is the most amplified PAK in breast cancer (in approximately 8% of breast cancers) while PAK4 amplification is frequent in uterine, ovarian and pancreatic cancers but is only detected in approximately 2% of breast tumors in the TCGA dataset (Kumar & Li, Advances in Cancer Research 2016).

Using the cBioPortal database (Cerami *et al.*, Cancer Discovery 2012; Gao *et al.*, Science Signaling 2013), we replicated this finding of PAK4 in the TCGA dataset and we also expanded

the analysis to the METABRIC breast cancer cohort where we find a comparable percentage of tumors with PAK4 amplification. Interestingly, patients carrying tumors that harbor PAK4 amplification tended to exhibit worse prognosis, suggesting that although not a very frequent alteration, PAK4 amplification may still be a clinically relevant feature. We display this new information in Supplementary Fig. S1f-h.

Additionally, we analyzed PAK4 copy number and mutational status in the breast cancer cell lines used throughout the study but no relevant alterations were found. This information is summarized in the new Supplementary Table 1.

In connection to the presentation of these data in Results, we have also clarified the difference in abundance of gene amplification of PAK1 and PAK4 in breast cancer referring to the study by Kumar & Li, *Advances in Cancer Research* 2016.

3) In Figs S2H and S2I, what is the explanation for the variation in PAK4 expression. Incomplete deletion of the floxed gene by CRE?

This is a very relevant question and we have now performed additional analyses of PAK4 and Cre expression to address this issue.

While several aspects might contribute to the variation of PAK4 expression observed in our mouse model with conditional PAK4 knockout in the mammary epithelium, the most likely explanation stems from the previously recognized stochastic nature of transgene expression, namely in the MMTV-Cre line here used (Wagner *et al.*, *Nucleic Acids Research* 1997). In our model, the expression of Cre recombinase and PyMT is not coupled within the exact same mammary epithelial cells, meaning that, stochastically, a mosaic pattern of transgene expression is expected, where some cells will co-express Cre and PyMT and other cells will only express Cre, PyMT or none of the transgenes (Wagner *et al.*, *Nucleic Acids Research* 1997; Pylayeva *et al.*, *JCI* 2009). Due to the strong nature of the PyMT oncogene, it is likely that cells where only PyMT is expressed may eventually contribute to subpopulations of tumor cells where PAK4 is not even subjected to Cre-mediated knockout.

To address this complexity, we have now included immunohistochemistry (IHC) for Cre in tumors that arose in mice of both genotypes and scored the Cre expression. A mosaic pattern of Cre expression was indeed present, particularly in PyMT;PAK4^{MEp^{-/-}} mice (Supplementary Fig. S2o-p). A mosaic pattern of PyMT expression is less relevant here because only PyMT-transformed cells would develop into tumors in animals of this age.

We initially detected PAK4 overexpression in PyMT-driven tumors by immunoblot (IB) (Supplementary Fig. S2d-f) and we have also showed that while there was an overall lower PAK4 expression in late stage mammary tumors harvested from PyMT;PAK4^{MEp^{-/-}} mice, there was detectable PAK4 expression remaining that varied within the group (Supplementary Fig. S2h-i). However, immunoblotting does not allow the evaluation of the spatial distribution of PAK4 expression in the tissues and the discernment of discrete subpopulations of tumor cells with differential PAK4 expression.

To address that, we have now established working conditions and a semi-quantitative scoring system to detect low and high *PAK4* mRNA levels in murine mammary tissues using *in situ* hybridization (ISH / RNAScope). With this technique, we again show that *PAK4* expression is consistently lower in the tissues harvested from PyMT;*PAK4*^{MEp-/-} mice as compared to control (Supplementary Fig S2j-k). In addition, we can now show an overall lower *PAK4* expression in early lesions compared to late stage PyMT-driven tumors and, more importantly, we can now appreciate substantial *PAK4* expression remaining specifically in tumor areas of PyMT;*PAK4*^{MEp-/-} tissues (Supplementary Fig. S2j-k). This indicates that *PAK4* was not completely depleted from the mammary tissues of PyMT;*PAK4*^{MEp-/-} mice and may therefore still contribute to tumorigenesis. Notably, this residual *PAK4* actually underscores the detected impact of *PAK4* gene depletion on mammary tumor development in our model.

In addition, we have also performed IHC with a new anti-*PAK4* antibody that we generated here for this purpose because we were not satisfied with the performance of existing anti-*PAK4* antibodies that we have previously tested. This antibody detects high levels of *PAK4* protein in tissue sections (Supplementary Fig. S2l-m). IHC results are in agreement with the remaining techniques (IB and ISH) used to evaluate *PAK4* expression in these murine tissues.

Reviewer #2, Expertise: senescence (Remarks to the Author):

This manuscript by Costa et al. is about p21-activated kinase (PAK) 4, which is known to be overexpressed in various types of cancer, including breast cancer. PAKs have been linked to cell proliferation and metastasis, as well as drug resistance in cancer.

From database analysis, the authors confirmed that PAK4 is upregulated in breast tumor patient samples and cell lines. Higher PAK4 expression level also correlated with worse prognosis. They generated two mouse models with altered PAK4 level: one is mammary-specific PAK4-transgenic mouse that developed mammary tumor with 25% penetrance, and another is PyMT-driven breast cancer mouse model (activates Ras and PI3K) combined with conditional PAK4 knock-out, in which tumorigenesis was slower and the survival time was extended. When PAK4 was knocked-down in cancer cell lines, senescence-like features were observed and cell proliferation was inhibited showing the anti-tumor growth effect of PAK4 inactivation.

PAK4 knock down correlated with NF- κ B target gene activation, and patient sample analysis revealed the reverse correlation between PAK4 expression and NF- κ B signature. Furthermore, they showed that PAK4 physically interacts with RELB, phosphorylate at a specific site, which in turn, decreases RELB–DNA binding and target gene transcription, including CEBP β .

Overall, this is a thorough study about PAK4, encompassing patient sample analysis, mouse model, as well as biochemical experiments to reveal novel mechanism of PAK4 action.

Particularly, the link between PAK4 and NF- κ B signaling which have been correlated, but not specifically shown, is now revealed in a very convincing manner.

However, some aspects need to be further clarified before this manuscript can be considered for the publication. Most of them are related to the expression level and activation of PAK4 and RELB, and PAK4's role in senescence. In addition, the analysis method of RNS sequencing data should be better clarified. See below.

We were happy to read the reviewer's positive comments. We are grateful for the comments as they helped us to clarify the raised points, which improved the manuscript.

Major concerns

1. Throughout the manuscript, I was wondering about the connection between the protein level and activation of PAK4. In patient samples, it is the increased expression that is relevant and in most of the experiments, only PAK4 expression level was modified and monitored. In the mechanical study in NF- κ B signaling, it is also not clear about the activation status of PAK4. Is activation by Ras not relevant for PAK4's role in cell proliferation and senescence? And how important is its role in Ras-induced senescence?

We agree that the relation between PAK4 expression levels and its activation status is an interesting question. However, in spite of a large number of previous studies on PAK4, this relation is not entirely clear. While PAK4 kinase activity can be activated, the PAK4 expression level appears to be limiting in many cases since overexpression of PAK4 has been sufficient in previous studies to drive cell proliferation, cell migration and transformation, regardless of the mutational status of key oncogenes such as Ras. We also do not wish to claim that Ras can activate PAK4, since we have not been able to find any evidence for this in the literature.

Importantly, it is also correct that PAK4 is commonly amplified and overexpressed in several cancer forms and in most cancer cell lines, while PAK4 point mutations in cancer are rare (Kumar & Li, *Advances in Cancer Research* 2016). In a tumor evolutionary perspective, this points to a role for PAK4 expression levels in cancer rather than a need for activation.

It should be noted that Ras-induced senescence occurs in untransformed cells that typically express very low PAK4 levels. This is also the case in the human mammary epithelial cells (HMECs) that we have examined in Fig. 3k, where H-RasV12 expression induced senescence in a context with barely detectable PAK4 expression (Supplementary Fig. S1d). This means that the induction of senescence by oncogenic Ras in this model most likely occurs independently of PAK4. However, by exogenous *wt* PAK4 expression, we were able to partially rescue HMEC cells from senescence, suggesting that PAK4 can protect untransformed cells from Ras-induced senescence. Also this Ras-contradictory function of PAK4 indicate that PAK4 does not likely operate downstream of Ras in induction of senescence.

Our previous studies in breast cancer cells show that exogenous expression of *wt* PAK4 promotes cell migration in a PAK4-kinase dependent manner (Zhang *et al.*, *JCB* 2002; Li *et al.*, *JBC* 2010). Similarly, in other studies, overexpression of *wt* PAK4 caused mouse fibroblasts to become tumorigenic *in vivo* (Liu *et al.*, *Molecular Cancer Research* 2008). Thus, cellular effects dependent on PAK4 kinase activity can be conveyed by PAK4 overexpression. This is similar to what is occurring in many cancer forms, which overexpress *wt* PAK4, but of which most lack Ras-mutations. Thus, *wt* PAK4 overexpression by itself correlates to cancer, to altered cellular functions as well as to increased PAK4 kinase activity. This means that the PAK4 expression level appears to be limiting for the PAK4 kinase activity and PAK4-induced phenotypes.

In our experience, we also detect a high endogenous PAK4 activity in cells (Zhang *et al.*, *JCB* 2002; Li *et al.*, *JBC* 2010). This is different from what we have observed for PAK1, where we failed to detect significant kinase activity without adding constitutively active small GTPases Rac1 or Cdc42 (Thullberg *et al.*, *Oncogene* 2007). PAK1 thus displayed a tighter auto-inhibitory regulation of its kinase activity compared to PAK4. This is also consistent with the previous finding that overexpression of a catalytically active mutant of PAK1 lead to the development of malignant mammary tumors in 20% of transgenic females and to a variety of other breast lesions (Wang *et al.*, *Oncogene* 2006). Thus, the relatively low penetrance and long latency phenotype of their mouse model overexpressing catalytically active PAK1 is comparable to the phenotype of our transgenic model overexpressing *wt* PAK4.

In further support of *wt* PAK4 carrying significant kinase activity without activation is our observation that recombinant full length *wt* PAK4 (used in previous studies as well as here for mechanistic experiments of NF- κ B/RELB) is highly kinase active without activation (e.g. Li *et al.*, *JBC* 2010; this paper Fig. 6b-c), meaning that at least a significant fraction of *wt* PAK4 exists in a kinase active conformation without the need for further activation.

Taken together, this indicates that PAK4 kinase activity and PAK4-induced phenotypes correlate to *wt* PAK4 expression levels.

2. (related to 1.) *Breast cancer cell lines have high PAK4 level. Do they also have Ras/Raf activation mutation? It looks like, only high level does not always cause tumor (see mouse model low penetrance). Therefore, it must require additional activation probably by Ras. Has it been tested? For example, by PAK4 overexpression in PyMT model? Or database analysis?*

We thank the reviewer for the suggestion to clarify the mutational status of the utilized breast cancer cell lines, which helps to clarify the oncogenic context of these cell models. This information is now summarized in the new Supplementary Table 1. The breast cancer cells here used varied regarding Ras/Raf. While all of the used breast cancer cell lines display high PAK4 expression (Supplementary Fig. S1d), these cells varied regarding the Ras/Raf copy number and mutational status, with some of these cell lines lacking alterations in Ras/Raf (Supplementary Table 1). Our finding that all these different cell lines all responded in a similar fashion upon PAK4 ablation indicates that the here found role of PAK4 is not linked to the Ras/Raf status.

Regarding the role of *wt* PAK4 expression and the potential relation between Ras and PAK4 activity, please see point 1 above.

As noted by the reviewer, MMTV-PAK4 overexpression in the mammary epithelium caused tumors only in a fraction of the mice. These tumors also appeared late in comparison to transgenic models with overexpression of strong oncogenes. As noted, this means that PAK4 overexpression may not be sufficient to drive mammary cancer. However, in the light of our finding that PAK4 overexpression can rescue mammary epithelial cells from oncogenic H-RASV12-induced senescence (Fig. 3k), PAK4 may have a facilitating role in tumor formation, rather than being a driver. In this role, overexpression of PAK4 would allow mammary epithelial cells to evade oncogene induced senescence upon RAS mutations, while cells without PAK4 overexpression may instead undergo senescence and therefore cannot form tumors. Our finding that a majority of the mammary tumors in MMTV-PAK4 transgenic mice displayed activating K-RAS mutations is consistent with such a facilitator role for PAK4. In an attempt to clarify this issue we have rephrased the results section and explained this further in the discussion of the revised manuscript.

As suggested by the reviewer, we have also crossed our mouse model of MMTV-PAK4 overexpression with MMTV-PyMT mice but we failed to detect an acceleration of tumorigenesis. This may be due to the short latency and extremely fast PyMT-tumorigenesis in the FVB/N background used for this experiment (Davie *et al.*, Transgenic Research 2007) as well as the high degree of heterogeneity in tumor development in this model, which may blur differences in tumor latency.

3. *In the PAK4 overexpression transgenic model, some of the mice developed spontaneous oncogenic K-Ras mutation, which is constitutively active, and in combined with increased PAK4 level, it would lead to more activated PAK4. Was there any consequence in tumor biology due to activated PAK4?*

As mentioned above, available data indicate that the amount of PAK4 kinase activity can be controlled by its *wt* expression level and we could find no evidence in the literature that Ras induces PAK4 kinase activity. Additionally, PAK4 expression levels are high across all the five

breast cancer cell lines here used (Supplementary Fig. S1d) independently of the Ras mutational status (new Supplementary Table 1).

As mentioned above under point 2, given the low penetrance and long latency of mammary tumors in the MMTV-PAK4 transgenic model, we do not suggest that PAK4 itself is a strong driver of cancer. The correlation with oncogenic Ras in the tumors of the transgenic model may instead suggest that PAK4 is a facilitator of tumor progression upon spontaneous activating Ras mutations. This is supported by the role for PAK4 overexpression in mitigating Ras-induced senescence as we here have shown in human mammary epithelial cells (Fig. 3k, please also see above). Thus, upon expression of mutant Ras, mammary epithelial cells that would normally become senescent, could continue to proliferate if PAK4 expression is high, and thereby these cells may form tumors driven by oncogenic Ras. We have tried to clarify these aspects in Discussion.

However, since the potential role of PAK4 overexpression for endogenous tumor development has not previously been examined, the role of its activity state during tumor development remains unclear. Resolving this will require transgenic expression of PAK4 mutants, which may constitute an interesting direction for the future. However, given that PAK4 is rarely mutated in human cancer, but is instead overexpressed, activating PAK4 mutants would not mimic the situation in human cancer.

4. Figure S2H: According to this result, some of PyMT;PAK4^{MEp}^{-/-} mice still retain PAK4 expression. Obviously, this is an issue when interpreting the Kaplan-Meier plot (Fig. 2F-H) as well as histological analysis (Fig. 2D and E). Have the authors consider stratifying the result in terms of PAK4 level in tumors? Was there any correlation between PAK4 level in tumor and the tumor onset or survival time?

We thank the reviewer for this suggestion. In our study, PyMT-driven tumors developed in all 10 murine mammary glands simultaneously; often multiple tumors per gland, until eventually, a faster/dominant tumor reached the humane endpoint determined by our ethical application (and used here as a proxy of overall survival). Because tumors were harvested for tissue analysis at sacrifice rather than at their first appearance, we analyzed the correlation of expression levels and survival.

We initially analyzed PAK4 expression in PyMT-driven tumors by immunoblot, but this methodology is not very quantitative and, as such, unsuitable for the suggested correlation analyses. To address this question raised by the reviewer, we evaluated PAK4 expression in the dominant tumors (that determined the sacrifice of the mice) using both *in situ* hybridization (ISH / RNAScope) and immunohistochemistry (IHC). The ISH technique allowed the spatial detection of high and low PAK4 mRNA expression and, as such, we could identify substantial PAK4 expression remaining specifically in tumor areas of PyMT;PAK4^{MEp}^{-/-} tissues (Supplementary Fig. S2j-k), indicating incomplete Cre-mediated PAK4 ablation, which is a well-established phenomenon in the MMTV-Cre model (Wagner *et al.*, Nucleic Acids Research 1997; Pylayeva *et al.*, JCI 2009).

In addition, we have performed IHC with an anti-PAK4 antibody that we generated here for this purpose because we were not satisfied with the performance of existing anti-PAK4 antibodies that we have previously tested. This antibody readily detected high levels of PAK4 protein in tissue sections (Supplementary Fig. S2l-m). IHC results are in agreement with the remaining techniques (IB and ISH) used to evaluate PAK4 expression in these murine tissues.

When plotting the PAK4 labelling score against survival time, the strongest correlation segregated the two genotypes of interest (Supplementary Fig. S2n). The strong divide in expression level between the two genotypes also represents the only obvious stratification based on PAK4 expression levels. This stratification was used in our Kaplan-Maier plots in Fig 2f-g. Importantly, this correlation is still apparent within the control group where PAK4 expression is overall substantial and therefore detected by the antibody. In the PyMT;PAK4^{MEp-/-} group, the range of PAK4 levels are below optimal detection levels and as such, unfit to own correlation analysis.

Notably, the finding that some tumors in the PyMT;PAK4^{MEp-/-} group express residual PAK4 actually underscores the effect of PAK4 ablation on tumor development in this *in vivo* model.

5. MMTV-PyMT is an aggressive metastatic breast cancer model. Considering the known function of PAK4 in cell proliferation and migration, it is plausible to expect that PyMT;PAK4MEp-/- mice would have less metastasis. Has this been addressed? Also, in relation with the PAK4 level (see comment above)?

We thank the reviewer for this interesting suggestion. As suggested, we have examined potential lung metastasis in our model, but we were unable to detect any lung metastasis in the control PyMT group within the time period during which we could maintain our mice as limited by our ethical permit.

It is well established that different genetic murine backgrounds have an impact on the resulting PyMT-driven phenotype (Davie *et al.*, Transgenic Research 2007). The original PyMT model in the FVB/N background (Guy *et al.*, Molecular and Cellular Biology 1992) is often used due to its short tumor latency and high incidence of pulmonary metastasis. However, the PyMT model obtained after continued backcrossing into the C57Bl/6J (B6) strain has a much longer latency. This strain seems to be less susceptible and more resistant to PyMT-driven mammary tumorigenesis (Davie *et al.*, Transgenic Research 2007). We have used the PyMT model in B6 background to match the strain background of our conditional PAK4 KO in order to study targeted disruption of PAK4 in mammary tumorigenesis. As noted above, we could not detect any metastasis in this model.

6. Figure 3C and S3C: how was the ‘senescence signature’ defined? Just ‘literature review’ doesn’t seem to be as strong and objective basis. To be clear, the direction of gene regulation – what is up or down in senescence – should be annotated and compared as two separate signatures. Is there overlap of senescence state and the genes upregulated in PAK4 knock down cells? Moreover, how much two cell lines Hs578T and BT-549 share this signature in the same direction is not mentioned. I am also not convinced about this strongly separating heatmap,

considering only part of the population entered senescence after almost complete knock down of PAK4 (e.g. ca. 50% cells are still BrdU-positive in all results, only a few cells are SA- β -gal positive in Fig S3E).

We agree with the reviewer that the used senescence signature lacked appropriate support. In recent studies (*i.e.* Hoare *et al.*, Nature Cell Biology 2016), time-resolved analyses of cells undergoing senescence have unveiled a dynamic pattern with various factors peaking at different phases of the senescence process. As such, analyses of a single, early time point, such as 72 hours post-transfection wouldn't be consistent with an established (late) senescence phenotype. In addition, most of the senescence signatures available in the literature have been generated in primary cells, mostly of fibroblast origin. Cancer cells that commonly have deficient p53/pRb/p16 status (as seen in the breast cancer cells herein used, see new Supplementary Table 1) would not be expected to display a signature that substantially overlaps with the classical primary cell-derived signatures. After careful consideration (discussed above) we think that such analyses in our set-up is not the most meaningful and we have therefore opted to remove these analyses from Fig. 3, Supplementary Fig. S3 and from the manuscript text.

7. Figures S3E – N: the result of 4-MU staining, implying senescence, is shown only fold induction compared to the control. To see the extent of senescence reaction, it would be more helpful to show how many cells in certain population is actually senescent. Therefore, I would recommend SA- β -gal staining or other method to show individual cells in or not in senescence.

We thank the reviewer for highlighting this point. Given our focus on breast cancer throughout our manuscript, we have addressed this question in all five human breast cancer cells utilized in this study. Firstly, we have transfected Hs 578T cells (a cell line that was extensively used throughout our study) with two independent siRNAs and we quantified the % of X-Gal positive cells at 4 and 7 days post-transfection. The results show that approximately 40% of the cells are X-Gal positive already 4 days post-transfection, with a slightly higher fraction of cells displaying SA- β -gal positivity after 7 days (Fig. 3a-c). This is in agreement with the percentage of BrdU-positive Hs 578T cells 5 days after transient PAK4 knockdown with 2 independent siRNAs (Fig. 3e) and with the fold-induction in SA- β -gal activity shown with the measurement of 4-MU fluorescence (Fig. 3d).

We have attempted to perform X-Gal staining as above in the other four breast cancer cell lines with limited success as some cells would display extremely high background (MCF7 and T-47D) and others would not retain any detectable staining (BT-549 and MDA-MB-231) upon our use of a standard X-gal staining protocol. We therefore analyzed the growth arrest at an individual cell level through quantification of EdU-incorporation in all of the breast cancer cell lines upon siPAK4 treatment (Supplementary Fig. S3h). These results are in agreement with the increased SA- β -Gal activity detected in cell extracts in all these cell lines by measuring 4-MU fluorescence (Fig. 3f).

8. Figure 3G-J: again, the impact of siPAK4 is only partial. To show that this is actually working, it would be important to show that siPAK4 cells, and only those with true knock-down

proliferate in presence of RAS. In addition, siPAK4 seems to have pro-proliferatory effect on the epithelial cells (J), therefore, it is also possible to postulate that the result is only the sum of increased (by siPAK4) and decreased (Ras) proliferation of the whole cell population, not by direct interaction of those two moieties. Cell sorting for double positive cells, and also for formal proof, PAK4 activation after Ras induction would be recommended.

Former Fig. 3g-j (now Fig. 3g-k) display data from two very different experimental setups but the placing of former Fig. 3j and the same color scheme may have made this unclear. We apologize for any confusion and we have now changed the color scheme of the latter to raise attention for the different experimental setups.

Fig. 3g-k show experiments of CRISPR/CAS9-mediated depletion of PAK4 in breast cancer cells, while Fig. 3k shows overexpression of *wt* PAK4 in untransformed human mammary epithelial cells that harbor inducible (4-OH-mediated) H-RASV12 expression.

The reviewer's comments appear to relate to the last experiment, shown in Fig. 3k, although this experiment does not involve any siRNA. All cells in this model are stably expressing 4-OH/tamoxifen inducible H-RASV12 expression resulting in RAS-induced senescence that is coupled to an almost complete block of cell proliferation (Borgdorff *et al.*, Oncogene 2010). We have transiently transfected these cells with EGFP or EGFP-PAK4 *wt*. In the control, EGFP-transfected population, H-RASV12 expression leads, as expected, to an almost complete growth arrest. Contrary, EGFP-PAK4 *wt* overexpression rescues the proliferation of a significant fraction of the H-RASV12-expressing cells. Only EGFP-positive cells (EGFP / EGFP-PAK4), as determined by FACS, were considered in this analysis. However, given that the used cells stably express inducible H-RasV12, meaning that all cells express H-RasV12 upon induction, as verified by the almost complete growth arrest, sorting based on H-RasV12 expression appears unnecessary. Further, as mentioned above, Ras does not likely cause senescence through activation of PAK4, since PAK4 levels in the used HMEC cells are barely detectable while Ras-induction gives almost complete growth arrest (Borgdorff *et al.*, Oncogene 2010 and Fig. 3k) and since PAK4 overexpression actually counteracted H-RasV12 by rescuing the proliferation from the Ras-induced complete growth arrest occurring in the absence of PAK4.

9. How fast is PAK4 in action? Cells with PAK4 CRISPR-CAS9 knock-down (Fig 3H) and xenograft of shPAK4 cells actually grow to a certain extent. Particularly for Xenograft, it seems they increase the tumor volume initially, although one assumes that the PAK4 was already inactivated at the time of inoculation.

In our *in vitro* experiments, exposure to recombinant PAK4 (without the need for co-activators) results in substantial substrate phosphorylation within minutes (Zhang *et al.*, JCB 2002; Li *et al.*, JCB 2010; this paper Fig. 6b-c). This suggests that PAK4 action is more or less instant.

In our hands, siRNA-mediated PAK4 depletion routinely takes over 48 hours to be detected by immunoblot, peaks at 4/5 days post-transfection and is still substantial at 7 days post-transfection (even though PAK4 levels slowly start resuming by then). As early as 72 hours post-transfection, siPAK4 cells exhibit a global expression profile that largely differs from that of control cells, with several differentially expressed genes (Supplementary Fig. S3a-d), suggesting that

consequences of PAK4 depletion start early on. However, it does take 4 to 7 days for the senescent-like phenotype to be gradually established in approximately 50% of the siPAK4-treated cell populations (as seen in SA- β -gal assays and BrdU/EdU-incorporation experiments in Fig. 3 and Supplementary Fig. S3). Likewise, cells treated with CRISPR/CAS9 PAK4-targeting guide sequences and cells stably expressing shPAK4 do grow to a certain extent. One possible explanation is that we cannot achieve complete PAK4 depletion with these set-ups. Remaining PAK4 expression in siPAK4 and shPAK4 cells can be appreciated in the immunoblots shown throughout Fig. 3, Supplementary Fig. S3 and in Supplementary Fig. S4a. Particularly for shPAK4 cells, our experience is that after long term culture we can no longer detect a difference in PAK4 expression compared to control cells as PAK4-expressing cells overgrow and take over this population. With CRISPR/CAS9 technology we seem to more efficiently deplete PAK4 as shown in Fig. 3j and this correlates to a more efficient growth arrest than what we observed when using RNAi (Fig 3 and Supplementary Fig. 3). However, while we can culture the CRISPR/CAS9 sgPAK4 cells for a limited time after infection/selection, we have been unable to expand them beyond a few passages.

One can also speculate that the use of these constitutive strategies for PAK4 depletion may have put additional pressure for cells to overcome a deficiency in PAK4 and may have given room to hypothetical cell sub-populations that grow independently of PAK4. This is a well-established phenomenon during cancer treatments and the development of resistance to cancer therapies.

10. Fig 5A and S5A: According to the legend, this is a clustered heatmap of 'NF- κ B target genes that are differentially expressed', of which some are upregulated upon siPAK4. However, about 50% of the genes shown here are downregulated. What are those, NF- κ B targets but regulated in significantly opposite direction upon PAK4 knock-down?

NF- κ B is a homo- or heterodimeric transcriptional complex formed by the 5 potential family members RELA, RELB, NF- κ B1, NF- κ B2 and c-REL. Different dimer combinations can act as transcriptional activators or repressors, as is known for other transcription factors (Zhang *et al.*, Cell 2017).

The utilized list of NF- κ B target genes was derived from a third party dataset based on genome-wide ChIP-seq analysis of RELA promoter binding (<https://www.ncbi.nlm.nih.gov/geo/query/acc.cgi?acc=GSM1055810>). The list therefore includes genes likely regulated by RELA-containing NF- κ B complexes, but the data is agnostic to the direction and degree of such regulation. We therefore cannot from these data differentiate which genes may be expected to be up- or downregulated. It is therefore not possible to try to match up- or downregulated genes in our datasets to an expected directional pattern in the NF- κ B target gene list. Nevertheless, we found a significant enrichment of NF- κ B target genes upon PAK4 siRNA treatment ($p < 0.001$) (Fig 5a and Supplementary Fig. S5a).

We have now clarified which are the NF- κ B target genes that are up- or downregulated by PAK4 siRNA in Supplementary Tables 8 and 9 for Hs 578T and BT-549 cells, respectively, and specified a selection of these directly in Fig 5a and Supplementary Fig. S5a.

11. Related to 10, the authors describe in the methods section that the list of NF- κ B target genes was derived from the public dataset (which is human sequences) and then the ‘raw sequences (RNA-seq data from HS578T?) were aligned against the mm9 mouse genome...’, which confused me further. Why did they use mouse genome? This has to be better clarified for readers who may not be an expert in RNA expression data analysis.

We thank the reviewer for carefully reading the manuscript and we apologize for any confusion this error may have caused. We have now corrected the Methods section as this should state “were aligned against the hg19 human genome”.

12. Fig 5E and S5E: The authors claim that PAK4 negatively regulate NF- κ B and senescence-like arrest. Therefore, when there is less PAK4 (siPAK4), there are more NF- κ B expressed, more p53, cells proliferate less (senescence?). Does this mean, more PAK4 leads to less Ras-induced senescence? But Ras activates PAK4. How can one solve this discrepancy?

In the set up shown in Fig. 3k, we have specifically tested the role of PAK4 in Ras-induced senescence. Upon overexpressing EGFP-PAK4 in untransformed mammary epithelial cells where we have induced expression of H-RASV12, we do observe less Ras-induced senescence. However, as pointed out above (points 1-3), oncogenic Ras most likely does not induce senescence in a fashion mediated by PAK4. However, we do not wish to suggest that Ras activates PAK4, since we could not find any evidence for such an assumption.

In addition, all breast cancer cell lines tested displayed high PAK4 expression levels (Supplementary Fig. S1d). These include cells with HRAS G12D mutation (Hs 578T cells), KRAS G13D mutation (MDA-MB-231), N-RAS amplification/H-Ras deletion (MCF7) while all other breast cancer cell lines herein used have *wt* RAS genes (see new Supplementary Table 1). Importantly, PAK4 depletion inhibited proliferation in a senescence-like fashion in all of these cell lines, showing that the role of PAK4 in keeping breast cancer cells out of senescence is not linked to oncogenic Ras. Please also see points 1-3 above where we discuss the potential relation between Ras and PAK4.

13. It seems that PAK4 regulates RELB transcription (RNA-seq result and RT-qPCR). However, the mechanism suggested in Fig 6 is about RELB inactivation by PAK4. Please clarify this difference.

We thank the reviewer for pointing out this issue. It is correct that NFKB1, NFKB2 and RELB mRNA expression levels were all increased by PAK4 siRNA. RELB is a known transcriptional target of both types of NF- κ B signaling, meaning that the increased levels of NFKB1, NFKB2 and RELB, as well as the activation of RELB could act in synergy to increase RELB levels upon PAK4 depletion. Thus, activation of RELB by PAK4 elimination likely acts in synergy with the increased expression levels of RELB. We have clarified this in the Discussion of the revised manuscript.

Minor comments

1. In the PyMT-induced breast tumor, what was the reason of PAK4 induction? Was amplification of the locus as found in patient samples?

We show in Fig. 1 and Supplementary Fig. S1 that there is significant PAK4 mRNA upregulation in human breast cancers and we have detected PAK4 protein overexpression in breast cancer cell lines and in PyMT-driven murine mammary tumors (Supplementary Fig. S2). PAK4 amplification, however, occurs only in a small percentage of human breast tumors (Kumar & Li, *Advances in Cancer Research* 2016; Supplementary Fig. S1f) and none of the breast cancer cell lines used show alterations in PAK4 copy number or mutational status (new Supplementary Table 1).

We also searched the literature and databases to investigate if PAK4 is amplified in the PyMT mouse model. Rennhack *et al.* have identified genomic copy number alterations in 600 tumors across 27 major mouse models of breast cancer, including the PyMT model here used. However, no amplification of PAK4 was detected in the PyMT model. In fact, MMTV-PyMT tumors have very few, if any amplicons (Rennhack *et al.*, *Journal of Mammary Gland Biology Neoplasia* 2017). Two older publications (Montagna *et al.*, *Molecular Biology and Genetics* 2003 and Hodgson *et al.*, *Molecular Biology, Pathobiology and Genetics* 2005) have used array comparative genomic hybridization analysis and essentially find only an amplicon at 11q (*Pak4* is at chromosome 7). Two more recent-papers (Ben-David *et al.*, *Nature Communications* 2016 and Rennhack *et al.*, *Journal of Mammary Gland Biology Neoplasia* 2017) use gene expression profiles to infer copy number aberrations and once again 11q is the only position that shows any signs of having an aberration while chromosome 7 does not display any deviations.

The PyMT oncoprotein binds to and co-opts several oncogenic signal transduction pathways, including those of the p53, *Src* family as well as the *ras* and PI3 kinase pathways (Lin *et al.*, *The American journal of Pathology* 2003), constituting candidates for causing the alteration of PAK4 expression in this model.

2. Figures 2F, G, and H: the number of mice shown here should be indicated.

As suggested, the sample size has been added to the legend.

3. Figure 2H: what is the meaning of this result? What kind of value does it add to the story?

Fig. 2h shows the PyMT-driven tumor onset in male mice, which can be considered as a distinct model of mammary tumorigenesis as compared to the PyMT-driven breast cancer model in females. While breast cancer is common in females and is universally studied in females, male breast cancer is an uncommon disease. Male breast cancer in human usually presents at later and more advanced stages and, as a consequence, survival rates are lower for men and have not improved over the recent years as female outcomes have (Anderson *et al.*, *JCO* 2010). Thus, there is an obvious importance of also studying male breast cancer. To this end, in Fig. 2h we

have studied the role of PAK4 in a mouse model system of male breast cancer. Although also PyMT-driven, biologically, this is a distinct mammary gland cancer model where the influence of female sex hormones is diminished, which results in a much slower kinetics of tumor progression. The large difference in mean tumor latencies in females and males allows us to better discern the differences in latency prompted by PAK4 abrogation. The finding that PAK4 plays an important role also in this model system further supports the notion of a general impact of PAK4 in mammary tumorigenesis.

4. Line 201 – 205: ‘Five breast cancer cell lines..... metastatic potential’. It would be helpful if this statement is supported by a supplementary figure or table. In addition, Ras/Raf mutational status should be also listed, since this is important for PAK4 activation. For their senescence potential, and particularly related to in vivo analysis, p53 mutational status should be also indicated for breast cancer cell lines.

We thank the reviewer for this suggestion. We have now added a new Supplementary Table 1 where all the requested information can be found. We have also added information regarding pRb and p16 as these are main senescence regulators. It is possible to see that the breast cancer cell lines vary with regards to p53 and Ras status suggesting that the role of PAK4 in senescence-like growth arrest is independent of these factors.

5. Line 205: Figure S3D is not a SA-β-gal staining result.

We thank the reviewer for carefully reading the manuscript. We have now corrected this in the revised manuscript.

6. Figure s3E: please quantify the result.

We have repeated and improved the analysis of X-Gal positivity upon transient PAK4 knockdown. The percentage of X-Gal positive cells has now been quantified at 4 and 7 days after transient transfection with two independent PAK-targeting siRNAs, and appropriate controls. The new data can be found in Fig. 3a-c.

7. Figure 3D-F: senescence is a terminal cell cycle arrest and it is important to show that cells are indeed arrested in G1. Please do cell cycle analysis of siPAK4 cells.

While we do not wish to contradict the reviewer, we found examples in the literature where senescence was associated with cell cycle arrest also in G2/M and that the nature of the arrest appears to be context dependent, with cancer cells becoming senescent through arrest in different phases of the cell cycle (reviewed by Campisi and d’Adda di Fagagna, Nature Reviews Cancer 2007). As suggested, we have analyzed cell cycle profiles of Hs 578T, MCF7 and BT-549 breast cancer cell lines upon siRNA or CRISPR/CAS9-mediated PAK4 depletion. Hs 578T cells appear to arrest in G2/M while MCF7 and BT-549 cells accumulated in G0/G1 upon PAK4 depletion. These data have been added to Supplementary Fig. S3m-o.

8. Figure 3G-H: similar to the comment above, please show G1 cell cycle arrest and SA- β -gal staining result to prove that this is senescence(-like) arrest.

We have now analyzed the cell cycle profile of BT-549 cells where PAK4 was ablated using CRISPR/CAS9 technology. These cells arrest in the G0/G1 phase of the cell cycle as seen in Supplementary Fig. S3o.

We have also attempted to perform X-Gal staining in these cells but, as discussed in the major point 7 above, BT-549 cells do not retain any detectable X-Gal staining after our use of a standard X-gal staining protocol. Instead, we have measured 4-MU fluorescence in cell extracts as an alternative marker of SA- β -gal activity and the results are presented in Fig. 3i.

9. Figure 4C: please indicate the sample number and also quantify the result.

Former Fig. 4c (new Fig. 4i) is a quantitative bar graph displaying the mean tumor weight and the individual data points per experimental condition. We have added the sample size to the figure legend. We have also added images of the X-gal labeling of the entire set of tumors and the corresponding quantification of X-Gal positive areas to Supplementary Fig. S4d-e).

REVIEWERS' COMMENTS:

Reviewer #1 (Remarks to the Author):

While the authors couldn't answer all my questions, I think they made a good faith effort to do so. The revised work is convincing and potentially important.

Reviewer #2 (Remarks to the Author):

The authors successfully answered all the questions raised and there is no more concerns left for this manuscript. I would like to praise their extensive effort to accommodate - sometimes - different views and improve the manuscript.

Response to reviewers comments

REVIEWERS' COMMENTS:

Reviewer #1 (Remarks to the Author):

While the authors couldn't answer all my questions, I think they made a good faith effort to do so. The revised work is convincing and potentially important.

Reviewer #2 (Remarks to the Author):

The authors successfully answered all the questions raised and there is no more concerns left for this manuscript. I would like to praise their extensive effort to accommodate - sometimes - different views and improve the manuscript.

We thank the reviewers. No action needed.